# Purkinje cells of the cerebellum control deceleration of tongue movements

**Paul Hage**, **Mohammad Amin Fakharian**, **Alden M. Shoup**, **Jay S. Pi**,
**Ehsan Sedaghat-Nejad**, **Simon P. Orozco**, **In Kyu Jang**, **Vivian Looi**,
**Hisham Y. Elseweifi**, **Nazanin Mohammadrezaei**, **Alexander N. Vasserman**,
**Toren Arginteanu**, **Reza Shadmehr** *

Laboratory for Computational Motor Control, Department of Biomedical Engineering, Johns Hopkins
School of Medicine, Baltimore, Maryland, United States of America

* shadmehr@jhu.edu

journal.pbio.3003110

for Clinical Brain Research and Werner
Reichardt Center for Integrative Neuroscience,
University of Tuebingen, GERMANY

**Peer Review History:** PLOS recognizes the
benefits of transparency in the peer review
process; therefore, we enable the publication
of all of the content of peer review and
author responses alongside final, published
articles. The editorial history of this article is
available here: https://doi.org/10.1371/journal.
pbio.3003110

## Abstract

We use our tongue much like our hands: to interact with objects and transport them. For
example, we use our hands to sense properties of objects and transport them in the
nearby space, and we use our tongue to sense properties of food morsels and transport
them through the oral cavity. But what does the cerebellum contribute to control of tongue
movements? Here, we trained head-fixed marmosets to make skillful tongue movements
to harvest food from small tubes that were placed at sharp angles to their mouth. We
identified the lingual regions of the cerebellar vermis and then measured the contribution
of each Purkinje cell (P-cell) to control of the tongue by relying on the brief but complete
suppression that they experienced following an input from the inferior olive. When a P-cell
was suppressed during protraction, the tongue's trajectory became hypermetric, and
when the suppression took place during retraction, the tongue's return to the mouth was
slowed. Both effects were amplified when two P-cells were simultaneously suppressed.
Moreover, these effects were present even when the pauses were not due to the climbing
fiber input. Therefore, suppression of P-cells in the lingual vermis disrupted the forces that
would normally decelerate the tongue as it approached the target. Notably, the population
simple spike activity peaked near deceleration onset when the movement required pre-
cision (aiming for a tube), but not when the movement was for the purpose of grooming.
Thus, the P-cells appeared to signal when to stop protrusion as the tongue approached its
target.

## Introduction

We use our tongue to shape the air and generate sounds in order to communicate, and we use
our tongue to evaluate food morsels and transport them through the oral cavity in order to
eat. These skillful acts involve coordination of over 100 muscles [1], producing movements
that are fundamental to our existence. Damage to the cerebellum disrupts these movements,
resulting in abnormal muscle activation patterns [2] that bear a resemblance to ataxia of the
arm [3]. However, life without a cerebellum in humans [4], or inactivation of the deep cere-
bellar nuclei in mice [5], do not eliminate tongue movements. Rather, the movements become

**Data availability statement:** All relevant data are available at https://osf.io/WDXU4/

**Funding:** The work was supported by grants from the National Institutes of Health (https://www.ninds.nih.gov/) (R01-EB028156 to RS, R37-NS128416 to RS) and the National Science Foundation (https://www.nsf.gov/) (CNS-1714623 to RS). The funders had no role in study design, data collection and analysis, decision to publish, or preparation of the manuscript.

**Competing interests:** The authors have declared that no competing interests exist.

**Abbreviations:** CIs, confidence intervals; CSs, complex spikes; ISI, inter-spike interval; P-cell, Purkinje cell; SSs, simple spikes.

inaccurate. For example, if the activities of Purkinje cells (P-cells) are disrupted via silencing of molecular layer interneurons, the tongue's trajectory becomes erratic and the subject is no longer able to efficiently harvest liquid rewards [6]. Thus, the role of the cerebellum in control of the tongue may be similar to its function during control of the limbs [7] and the eyes [8,9]: stopping the movement on target. But how might the cerebellum achieve this?

In primates, stimulation of the fastigial nucleus moves the tongue predominantly in the ventral-dorsal axis, while stimulation of the dentate nucleus moves it mainly in the medial-lateral axis [10]. Notably, tongue muscles are most readily activated via stimulation of the fastigial nucleus (as compared to the other cerebellar nuclei) [11], suggesting that the P-cells in the vermis play a prominent role in control of the tongue. Unfortunately, there are no reports of P-cell activity in the vermis during targeted tongue movements in any species, but more is known regarding activities of P-cells in Crus I and Crus II (in rodents). For example, as a licking bout is about to start, many P-cells in Crus I and Crus II increase their simple spikes (SSs), while a smaller number exhibits a decrease [12]. Once the licking begins, the SS rates as a population are phase-locked to the rhythm of the lick, with peaks occurring near lick onset [5]. Complex spikes (CSs) also exhibit their highest rates during protraction [12,13]. But what is the relationship between the activities of P-cells and control of the tongue?

To answer this question, we sought an animal model that had a long tongue and could skillfully direct it to small targets. Marmosets are an attractive choice because they have a 21 mm tongue which they use to burrow into small holes and retrieve insects and sap [14]. Indeed, they have an extraordinary ability to control their tongue, vocalizing in order to label other marmosets during 2-way communication [15].

As we trained head-fixed marmosets to make saccades to visual targets and then rewarded them with food [16], we noticed that they could naturally bend and twist their tongue in order to insert it into small tubes, even when the tubes were placed at 90° with respect to their mouth [17]. Because their harvest was difficult, they chose to do many saccade trials, allowing the food to accumulate, then stopped working and claimed their cache by scooping the food out of the tube [17].

To quantify how the cerebellum was contributing to the control of the tongue, we recorded from tongue modulated P-cells in lobule VI and VII of the vermis. Then, we relied on the fact that the inferior olive not only transmitted unexpected sensory events to the cerebellum [8,18–20], it also acted as a stochastic perturbation that completely suppressed the P-cells [21,22], which then resulted in a small movement [23], or a disruption of the ongoing movement [24–26]. Using spike-triggered averaging on the climbing fiber input, we found that the resulting SS suppression altered the deceleration phase as the tongue approached the target, producing hypermetria.

This hypermetria was replicated when the P-cells experienced a long period of SS pause [27,28] without a preceding CS. That is, both a CS-induced SS suppression, and a long SS pause independently had the same effect on behavior: producing downstream forces that extended the tongue. Because as a population, the SS rates were greatest during the deceleration phase of protraction, the results suggested that the P-cells signaled downstream structures to stop the movement as the tongue approached its target. Indeed, this strong engagement of the P-cells was present when the tongue was aiming for a small tube, but not when the movement's purpose was to groom the face.

## Results

We trained marmosets (*n* = 3) to perform visually guided saccades in exchange for food (Fig 1A). The subjects performed a sequence of task-relevant saccades, at the end of which we delivered an increment of food (slurry mixture of apple sauce and monkey chow). This food

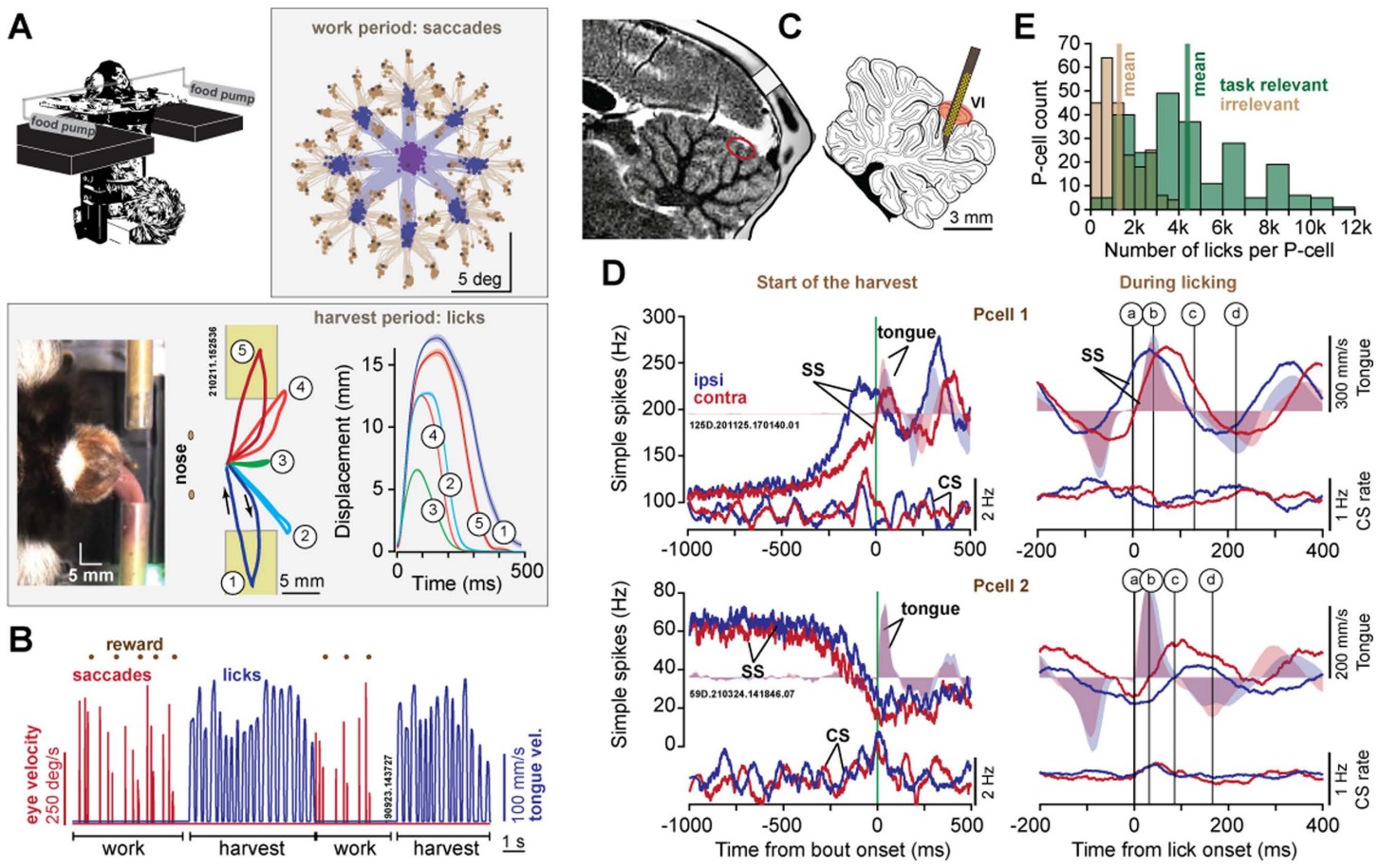

**Fig 1. Marmosets produced dexterous tongue movements during recordings from the cerebellar vermis. A.** Subjects made saccades to visual targets and received a small amount of food as reward via one of two tubes placed obliquely to the mouth. In the task-relevant licks, they directed their tongue to the edge of the tube to harvest food near the tip (trajectories 2, 4), or inside the tube to harvest food that was deeper (trajectories 1, 5). In task irrelevant licks, they groomed their face (trajectory 3). **B.** Subjects chose to work for consecutive trials, making saccades and allowing the food to accumulate, then harvested their cache in bouts of licking. **C.** We employed silicon probes to record from lobule VI and VII of the vermis. **D.** Simple- and complex spikes (SSs, CSs) of two Purkinje cells, aligned to bout onset and lick onset. A single lick was divided into acceleration period of protraction (a, b), deceleration period of protraction (b, c), and acceleration period of retraction (c, d). Filled color regions indicate tongue velocity. **E.** The number of task relevant (tube directed) and task irrelevant (grooming) licks recorded per neuron. The data underlying this figure can be found in https://osf.io/wdxu4/files/osfstorage.

was presented via either the left or the right tube for 50–300 consecutive trials, then switched tubes. Because the food amounts were small (0.015–0.02 mL), and the tubes were located at ±90° with respect to the mouth, the harvest was effortful [17], requiring skillful movements toward a target that was just large enough to accommodate the tongue (4.4 mm diameter tube). As a result, the subjects chose to work for a few consecutive saccade trials ($n = 6.1 \pm 0.02$ successful trials per work period), allowing the food to accumulate, then stopped making saccades to targets, fixated the tube and harvest via a bout of licking ($n = 22.03 \pm 0.04$ licks per harvest period, Fig 1B).

We tracked the motion of the tongue in the horizontal plane using DeepLabCut [29]. The tongue movements were of two general types: in the task-relevant licks the subjects aimed for the tube (S1 Video), whereas in the task-irrelevant licks the subjects groomed their mouth (Fig 1A) (S2, S3, and S4 Videos). During the 2- to 3-h recording sessions the subjects performed $n = 4{,}401 \pm 11$ task-relevant licks (i.e., aimed at a tube, mean ± SEM), and $n = 1{,}310 \pm 4$ task-irrelevant licks (Fig 1E).

Typically, the subjects began their harvest by licking the food near the tip of the tube (S5 Video), but then as the food cache declined, they inserted their tongue into the tube (S6 and S7 Videos, scooping out their reward. Thus, we divided the task-relevant licks into two subtypes, those that aimed for the edge of the tube and harvested the food that was near the tip (Fig 1A, labeled 2 & 4), and those that penetrated the tube and harvested the food that was deeper (Fig 1A, labeled 1 & 5). Lick protraction velocity was largest for inner tube licks, which also had the largest amplitude and longest protraction duration (S1 Fig). Duration of the protraction phase of the inner tube licks was longer than the duration of retraction. For example, in subject 132F, inner tube licks had a protraction duration of 201.3 ± 0.16 ms versus retraction duration of 133.6 ± 0.12 ms (S1 Fig). This is consistent with the idea that in contrast to retraction, protraction required aiming which tended to accompany longer-duration movements.

## Climbing fibers were most active near lick onset

We combined MRI and CT image-guided procedures [16] to place heptodes and silicon probes in lobules VI and VII of the vermis. Over the course of 3.5 years, we recorded from $n$ = 284 P-cells (Figs 1C and 1E) while the subjects performed 840,787 licks. A neuron was identified as a definitive P-cell ($n$ = 230) because of the presence of CSs. In addition, we included data from putative P-cells ($n$ = 54) for which we could not isolate the CSs, but the neuron was located in the P-cell layer and exhibited 0 ms synchronous SS interactions with other confirmed P-cells [8,9,26,30] (S2A Fig).

Among our P-cells, the SS modulations were usually present for both tongue movements and eye movements (S2C Fig). However, as our aim was to record from the P-cells that were tongue modulated, in our population the P-cells were more strongly modulated by tongue movements (paired t-test, $t$(156) = 7.96, $p$=3.3E−13). This preferential encoding of tongue versus eye was greater for neurons that were located in lobule VI (S2B Fig).

Fig 1D illustrates the activities of two P-cells near bout onset, as well as during licking. As the bout began, one P-cell increased its SS activity, earlier when the tongue targeted the ipsilateral tube (with respect to the site of recording), while another P-cell decreased its SS activity, earlier for the contralateral tube. As the licking continued, the SS rates in both P-cells were modulated in a rhythmic pattern. We focused on three periods during each lick: protraction acceleration period (Fig 1D, a, b), protraction deceleration period (b, c), and retraction acceleration period (c, d).

As a population, the $n$ = 230 confirmed P-cells exhibited CS rates that increased near the onset of protraction (Figs 2A), peaking around the time the tongue touched the tube, and then decreased below baseline around retraction onset. The increased rates during protraction were larger for ipsilateral licks (within cell difference, protraction period, mean ± SEM: 0.077 ± 0.024, $t$(229) = 26.7, $p$ = 1.9E−72). Thus, as a population, for both target directions, the phase of movement for which the CS rates were maximum was protraction (termed CS-on phase).

## P-cell suppression produced overshooting during protraction and slowed return during retraction

The climbing fiber input suppressed SS production (Fig 2B, left: all P-cells, right: single P-cell), lasting an average of 14.8 ± 0.44 ms during licking (time to 85% recovery of SS rate, mean ± SEM). However, the CS events were rare: a CS occurred in only 5.45 ± 0.01% of the licks during protraction acceleration period (mean ± SEM), 9.28 ± 0.01% of the licks during protraction deceleration period, and 6.59 ± 0.009% of the licks during retraction acceleration

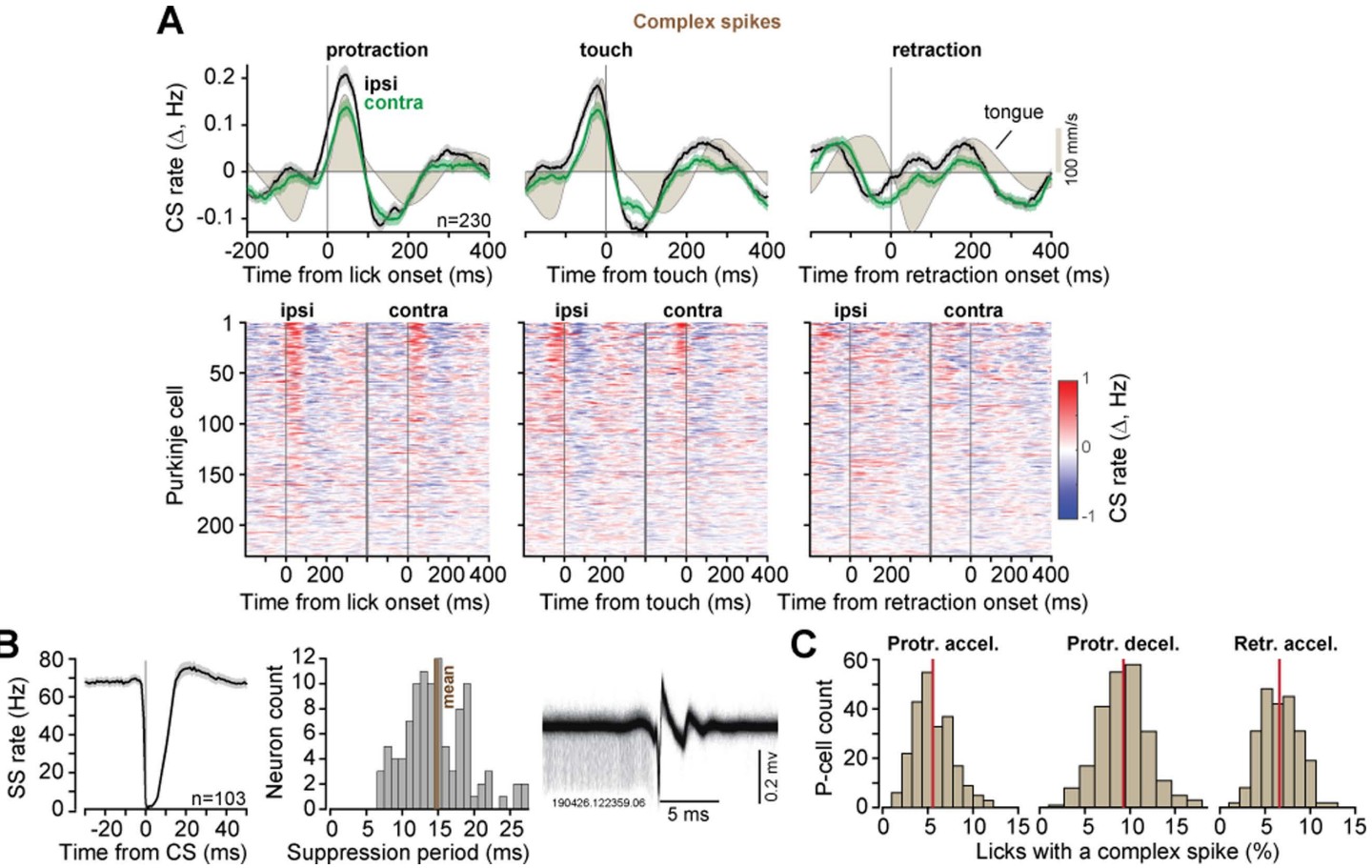

**Fig 2. CS rates increased with protraction and decreased with retraction. A**. CS rates across the population aligned to protraction onset, touch of the tube, and retraction onset. The second row shows CS activity across all P-cells, aligned to protraction, touch, and retraction, sorted based on CS rate for ipsilateral licks at lick onset. **B**. CS-induced SS suppression, averaged across the subset of P-cells for which both the CS and the SS were isolated (left). Examples of SS and CS waveforms for a single P-cell are shown at right. **C**. Percentage of licks with a CS during a specific period of time for each neuron. Vertical line indicates mean. Error bars are SEM. The data underlying this figure can be found in https://osf.io/wdxu4/files/osfstorage.

period (Fig 2C). We collected a large number of licks per neuron (4,401 ± 11 licks, Fig 1E), then performed CS-triggered averaging to ask whether the resulting SS suppression affected the motion of the tongue.

For each P-cell we considered triplets of consecutive licks $\{n-1, n, n+1\}$ in which all three licks were of the same type, i.e., contacted the same part of the tube (edge or inner). We then selected those triplets in which there was a CS at only a single period in lick $n$, but no CS during any period in the two neighboring licks $n-1$, and $n+1$. For example, consider licks in which there was a CS in one P-cell during the acceleration period of protrusion (S3 Fig). This acceleration period was brief (49.9 ± 0.019 ms), during which the SS activity was normally rising and nearly identical in licks $n-1$ and $n+1$ but suppressed during lick $n$ (S3 Fig, 1st row). We measured the change in SS activity by comparing lick $n$ with licks $n-1$ and $n+1$ and plotted the results of each comparison in S3B Fig, green and blue solid lines (labeled suppressed, top row, right column). As a control, we performed a bootstrapping procedure in which we generated a pseudo data set for each P-cell by randomly assigning the CS label to a lick and comparing it with its two temporally adjacent neighbors.

To ask whether there were any effects of the brief SS suppression on the tongue, we measured the distance of the tongue tip to the mouth and also its angle with respect to the midline, then compared the trajectories in lick $n$ with the neighboring licks in which the P-cell was not suppressed. The effects appeared consistent (S3A Fig, red colors): there was little or no change in tongue kinematics if the SS suppression occurred during the acceleration phase of protraction. Statistical testing, which relied on bootstrapping procedure to compute 95% confidence intervals (CIs), demonstrated that the changes were within the error bounds. Thus, the suppression during the acceleration period of protraction had no significant effects on tongue trajectory (S3C Fig, measured via distance to the mouth, and its angle, at peak protraction speed and peak displacement).

As the protraction continued, the effects of SS suppression became evident. If the suppression occurred during the deceleration period of protraction (duration: 103.6 ± 0.04 ms), the tongue exhibited hypermetria (Fig 3A, second row), producing increased displacement and increased angle of the tongue's trajectory (Fig 3B, displacement: 0.37 ± 0.002 mm, 95%CI = [−0.17, 0.14] mm, angle: 3.35 ± 0.02°, 95%CI = [−1.14, 0.98]°). Notably, the effects were consistent regardless of whether lick $n$ was compared to the previous lick ($n-1$), or the subsequent lick ($n+1$) (Fig 3B, second row, blue and green traces). Furthermore, the effects were consistent across the P-cells (Fig 3A, red colors, also S4 Fig, left panel), and larger for contralateral licks (within cell difference, ipsilateral minus contralateral, at lick endpoint, displacement mean ± SEM: −0.15 ± 0.04 mm, $t(229) = -3.97$, $p$ = 9.6E−05, angle: −1.41 ± 0.37, $t(229) = -3.79$, $p$ = 9.69E−05).

These results hinted that SS suppression prevented normal deceleration, producing hypermetria and a bending of the tongue away from the midline. If this interpretation is valid, then a similar suppression during the retraction period should produce forces that are again in the direction of protraction, now resisting the tongue's return. That is, if SS suppression during protraction sped the movement outward, then the same suppression during retraction should now slow the movement. In both cases, the suppression should bend the tongue away from the midline.

As retraction began, the population CS activity (Fig 2A) had fallen below baseline, i.e., opposite of the activity during protraction. Yet, if the CS occurred during the retraction acceleration period (duration: 78.3 ± 0.02 ms), the resulting suppression was again an outward displacement of the tongue and bending (Fig 3D). A comparison of lick $n$ with $n-1$ or $n+1$ revealed a consistent effect: an increased distance of the tongue to the mouth and an increased angle (Fig 3F, displacement: 1.29 ± 0.002 mm, 95%CI = [−0.32, 0.22] mm, angle: 4.70 ± 0.02°, 95%CI = [−1.15, 0.85]°). These effects were present for ipsilateral and contralateral licks (Fig 3E), larger for the contralateral licks (within cell difference, ipsilateral minus contralateral, at lick endpoint, displacement: −0.37 ± 0.09 mm, $t(229) = -3.97$, $p$ = 4.8E−05, angle: −2.06 ± 0.50, $t(229) = -4.1$, $p$ = 2.71E−05), and consistent across the P-cells (Fig 3D, red colors). Thus, the CS-induced SS suppression during retraction slowed the return of the tongue and producing bending away from the midline.

In summary, when the climbing fiber input briefly suppressed the P-cells during the deceleration period of the tongue's protrusion, the tongue overextended and bent away from the midline. When this suppression occurred during retraction, the tongue's return was slowed and again bent away from the midline. These effects were present for targets on both the ipsilateral and contralateral sides, but greater when the target was contralateral. Thus, it appeared that SS suppression disrupted the motor commands that would normally stop the tongue during protraction and return it during retraction. That is, the downstream effect of suppression of P-cells was to produce forces that extended the tongue and produced lateral bending.

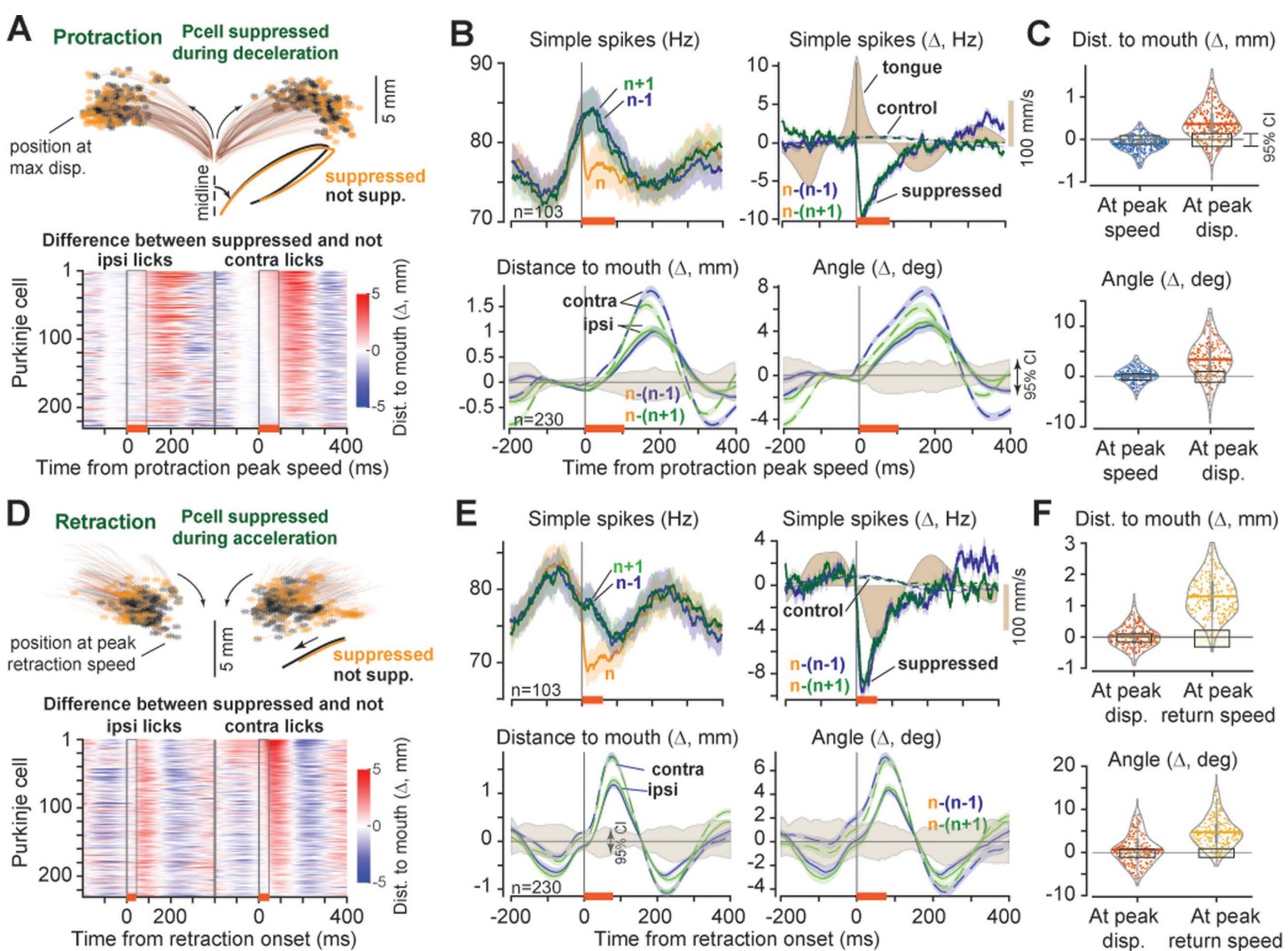

**Fig 3. CS-induced SS suppression produced hypermetria during protraction and slowing during retraction. A**. Suppression took place during the deceleration period of protraction. Traces show average tongue trajectory during this period of protraction for each P-cell during SS suppressed and control licks. Ipsilateral licks are shown to the left and contralateral to the right. Heatmap quantifies change in endpoint trajectory between suppressed and control licks for each cell. Period of suppression is indicated by the orange bar at the bottom of the heatmap. **B**. Top row: SS rates for licks $\{n-1, n, n+1\}$, where only lick $n$ experienced a CS. Filled color curves indicate tongue velocity. Second row: trajectory of the tongue in lick $n$ as compared to its two temporally neighboring licks. Trajectory is measured via distance from tip of the tongue to the mouth and angle of the tip with respect to midline. The filled region is 95%CI. **C**. Distance to mouth and angle in lick $n$ as compared to neighboring licks. Shaded region is 95%CI. **D-F**. Suppression during the acceleration period of retraction induced slowing. Same format as in parts A–C. In part F, tongue trajectory (displacement and angle) is similar before SS suppression (at peak displacement), but diverges after the suppression at peak return speed. Error bars are SEM. The data underlying this figure can be found in https://osf.io/wdxu4/files/osfstorage.

## Hypermetria increased when pairs of P-cells were simultaneously suppressed

Our dataset included $n$ = 298 pairs of simultaneously recorded P-cells. This allowed us to test whether near simultaneous suppression of two P-cells had a greater effect on tongue kinematics as compared to when only one of the two P-cells was suppressed.

As before, we collected triplets of consecutive licks $\{n-1, n, n+1\}$ where all the licks were of the same type (directed toward the same part of the tube), but only lick $n$ had a CS. We divided the triplets based on whether a CS was present in only one of the P-cells, or both

P-cells, then computed trajectory differences between licks $n$ and $n-1$, as well as licks $n$ and $n+1$. Finally, for each pair of P-cells we averaged $n-(n-1)$ and $n-(n+1)$ to increase statistical power (as there were far fewer licks in which both P-cells experienced a CS during the same period of the movement).

We found that if two P-cells were suppressed during the deceleration phase of protraction, then there was significantly greater displacement of the tongue, and bending, as compared to when only one of the P-cells was suppressed (Fig 4A, angle: $t(248) = -3.7167$, $p = 2.5E-04$, displacement: $t(248) = -5.5471$, $p = 7.43E-08$). Similarly, when the suppression occurred during retraction, the return phase of the movement experienced a greater slowing in the case of two P-cells as compared to a single P-cell (Fig 4B, angle: $t(192) = 6.34$, $p = 1.6E-9$, displacement: $t(192) = 6.50$, $p = 6.79E-10$). Thus, near simultaneous suppression of two P-cells roughly doubled the kinematic effects.

## Control studies

The fact that a CS was present during a given period in lick $n$ may have been because earlier in the tongue's trajectory there was an event (for example, an error), that affected that movement, increasing the likelihood of a CS, and resulting in compensatory movements that followed the CS. To check for this, for each period during which we observed a CS we considered the tongue's trajectory in the same lick but during the period preceding the CS. For example, for the licks in which there was a CS in the protraction deceleration period we focused on the acceleration period of the same movement. By comparing the lick in which the CS had occurred with its two neighbors, we found that in the period before the CS had occurred tongue kinematics remained within the 95% CIs of chance: distance to mouth and angle of the tongue at peak speed were not different than chance (Fig 3C, period before peak

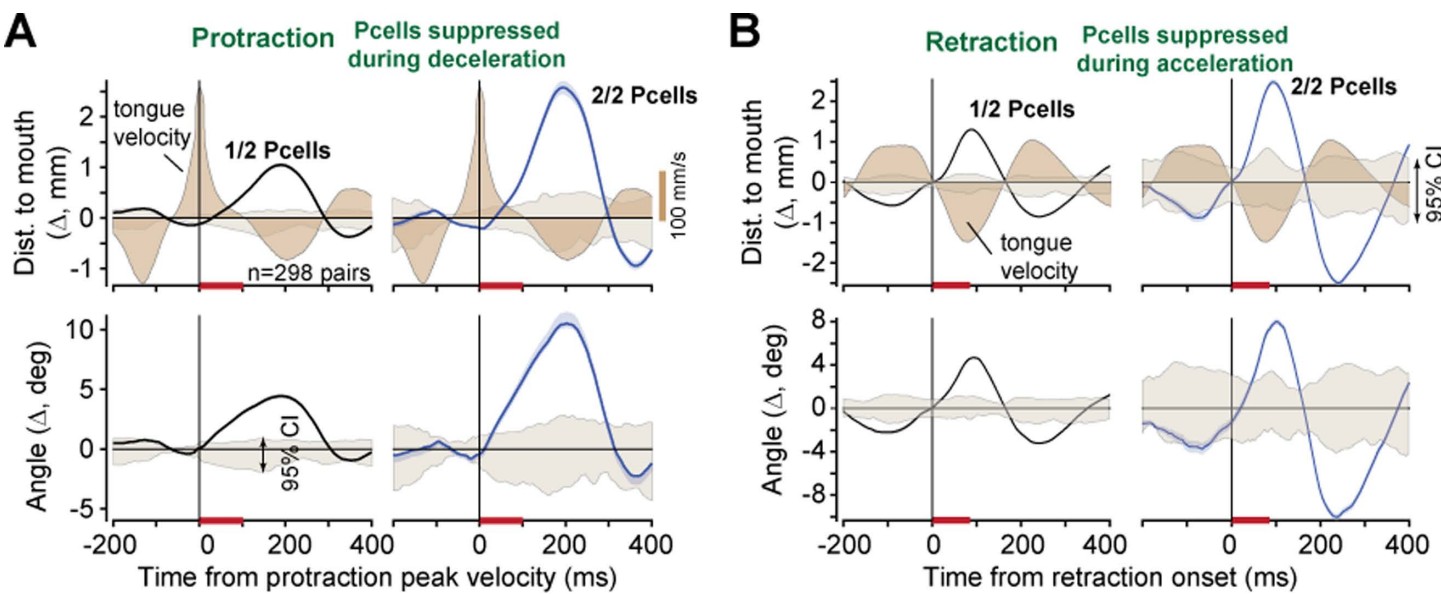

**Fig 4.. Suppression of multiple P-cells scaled the perturbation to the tongue. A.** Kinematic effects of CS-induced P-cell suppression during the deceleration period of protraction (orange bar). Traces show a change in tongue trajectory in suppressed licks vs. control licks, measured via distance from the tip of the tongue to the mouth and angle of the tip with respect to the midline. Gray shaded region is 95%CI. Brown-filled region is tongue velocity. **B.** Kinematic effects of P-cell suppression during the acceleration period of retraction (orange bar). Same format as in part A. Error bars are SEM. The data underlying this figure can be found in https://osf.io/wdxu4/files/osfstorage.

speed, displacement: −0.11 ± 0.001 mm, 95%CI = [−0.12, 0.09] mm, angle: 0.19 ± 0.01°, 95%CI = [−0.73, 0.53] °).

Next, we checked the licks in which there was a CS in the retraction period. We found that before the CS had occurred, at the onset of retraction the distance to the mouth and tongue angle were not different than if the CS had not occurred (Fig 3F, protraction period before peak displacement, displacement: 0.01 ± 0.002 mm, 95%CI = [−0.16, 0.10] mm, angle: 0.68 ± 0.02°, 95%CI = [−1.14, 0.83] °). Moreover, during the preceding protraction in the same lick, at peak tongue speed, the distance and angle were again not different than chance (displacement: −0.66 ± 0.003 mm, 95%CI = [−0.70, 0.74] mm, angle: −2.16 ± 0.02°, 95%CI = [−3.35, 3.58] °). Yet, if during the return phase a CS was present, the movement was slowed (Fig 3F, before peak-speed period of retraction, displacement: 1.29 ± 0.002 mm, 95%CI = [−0.32, 0.22] mm, angle: 4.70 ± 0.02°, 95%CI = [−1.16, 0.85] °). That is, the tongue's trajectory before the CS was not significantly different than the neighboring licks, but after the CS-triggered SS suppression the trajectories diverged.

As a further control, we considered movements in which there was a CS event just before protrusion onset. In these movements, the SS rates were suppressed, but the rates recovered around 10 ms after lick onset (S5 Fig, first row). Thus, despite the presence of a CS just before the lick had started, the SS rates during the entire protrusion and retraction periods were

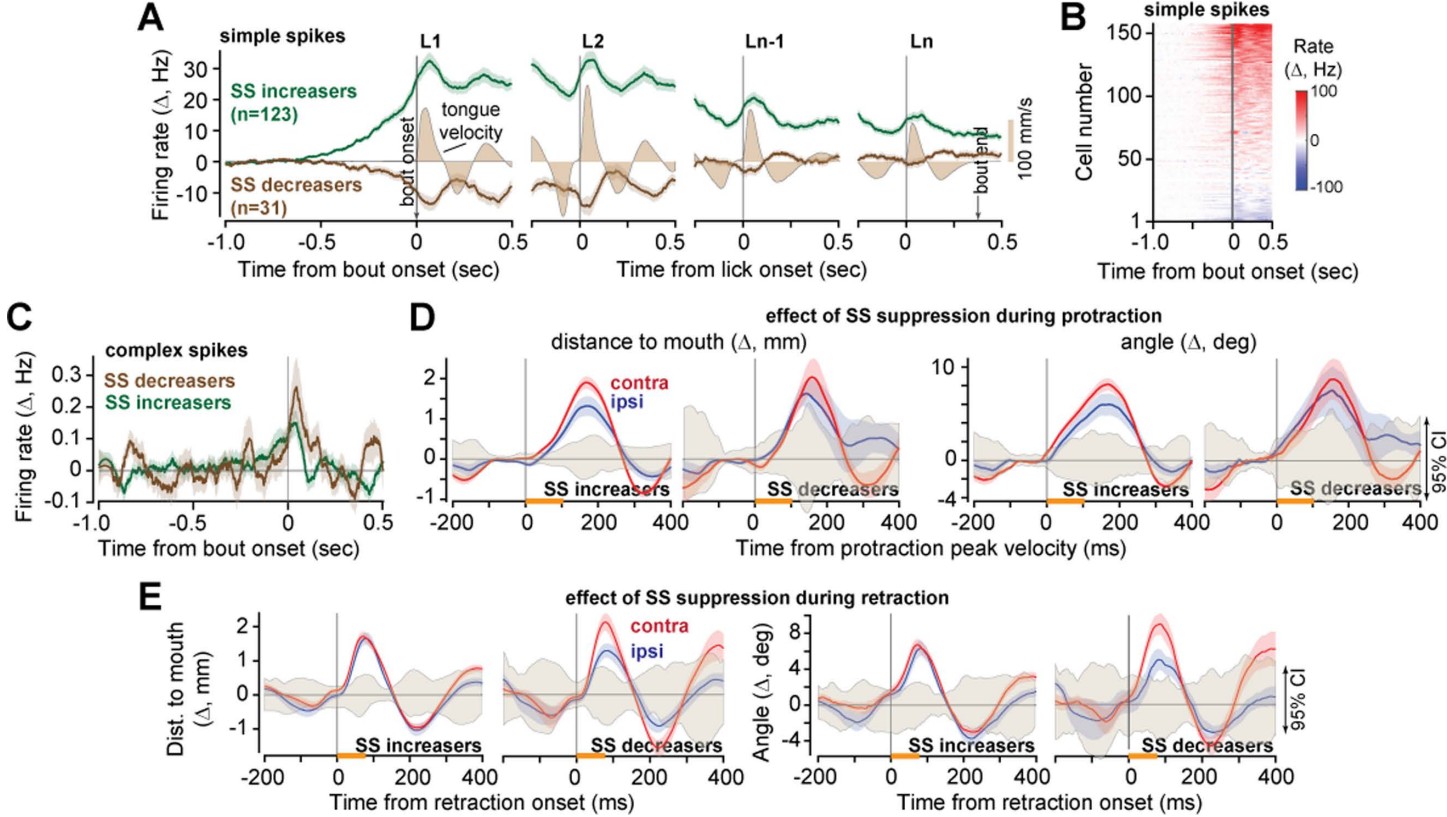

**Fig 5. Effects of SS suppression remained consistent across P-cells regardless of SS modulation. A.** Some of the P-cells exhibited an increase in SS rates before bout onset, while others exhibited a decrease. L1 is first lick in the bout, Ln is last lick. **B.** Change in SS rates for all P-cells (with respect to baseline), aligned to bout onset. **C.** Change in complex spike rate for the same P-cells. **D.** Kinematic effects of SS suppression during the deceleration period of protraction (orange bar). Shaded region is 95%CI. **E.** Kinematic effects of SS suppression during the acceleration period of retraction (orange bar). Error bars are SEM. The data underlying this figure can be found in https://osf.io/wdxu4/files/osfstorage.

intact. The trajectory of the tongue as measured via distance and angle remained within the 95% CI bounds of chance (S5 Fig, third and fourth rows).

## Effect of SS suppression remained consistent across P-cells regardless of SS modulation

For nearly every P-cell in our dataset, during both protraction and retraction, the CS-triggered SS suppression was followed by an extension of the tongue and a lateral bending (Fig 3A and 3D). This consistency was surprising in light of the diversity that was present in the SS patterns: before the onset of the bout, some P-cells had increased their SS rates with respect to baseline, while others had decreased (Fig 1D). Did the effects of SS suppression differ in these two groups?

In our data set of $n = 142$ of P-cells with SSs, most cells ($n = 123$) increased their SS rates before the onset of the bout, but a minority exhibited a decrease ($n = 31$) (Fig 5A and 5B). We separated the P-cells into SS increasers and decreasers and found that despite the differences in their SS patterns, the CS rates increased in both groups near bout onset (Fig 5C, peak CS firing rate change from baseline, SS increasers: 0.15 ± 0.04 Hz, SS decreasers: 0.26 ±0.10 Hz). We then compared tongue trajectories in triplets of consecutive licks $\{n-1, n, n+1\}$, finding that during protraction, following SS suppression in the deceleration period, in both groups of P-cells the tongue was displaced away from the mouth, exhibiting a greater distance and angle (Fig 5D, SS increasers, displacement: 0.38±0.005 mm, 95%CI = [−0.02, 0.02] mm, angle: 3.72 ± 0.05°, 95%CI = [−0.16, 0.15] °, SS decreasers: displacement: 0.45 ± 0.04 mm, 95%CI = [0.04–0.06] mm, angle: 4.28 ± 0.54°, 95%CI = [−0.16, 0.15] °). Similarly, during retraction, SS suppression in both groups of P-cells produced a slowing of the tongue's return, resulting in a greater distance and angle (Fig 5E, SS increasers, displacement: 1.25 ± 0.01 mm, 95%CI = [−0.04, 0.03] mm, angle: 4.45 ± 0.05°, 95%CI = [−0.16, 0.17] °, SS decreasers: displacement: 1.04 ± 0.04 mm, 95%CI = [−0.04, 0.03] mm, angle: 3.48 ± 0.17°, 95%CI = [−0.16, 0.17] °).

Thus, regardless of the patterns of SS activity during licking among the various P-cells, the downstream effect of CS-triggered SS suppression was extension of the tongue coupled with lateral bending.

## SS pause without a CS was sufficient to produce hypermetria

We had interpreted the kinematic effects that followed the CS-induced SS suppression as being caused by SS suppression, not due to the presence of a CS. To check the validity of this conjecture, we quantified the kinematic effects of non-CS-induced long SS pauses on the tongue's trajectory. To identify a long pause, for each P-cell we considered all licks towards the same direction in which no CS occurred at any point in the movement. We then identified the longest inter-spike interval (ISI) for the SSs that originated in each phase of each lick (phase refers to protraction deceleration period, etc.). For each phase under study, and each P-cell, we labeled 25% of the licks with the largest ISIs as a "long-pause" lick.

Next, we considered triplets of consecutive licks $\{n-1, n, n+1\}$ of the same type in the same direction in which none of the licks had a CS during any phase of the movement. We selected the subset of triplets in which lick $n$ had a long pause in only one phase of the movement, but no long pauses in any phase of the neighboring licks. We then compared tongue trajectories between the lick that had a long SS pause with its two neighboring licks.

On average, the duration of a long SS pause was 31.25 ± 1.2 ms during protraction deceleration, and 34.25 ± 1.08 ms (mean ± SEM) during retraction acceleration. If this pause occurred during the protraction acceleration period, it produced hypermetria and bending of the tongue away from the midline (Fig 6A), and if it occurred during the retraction period

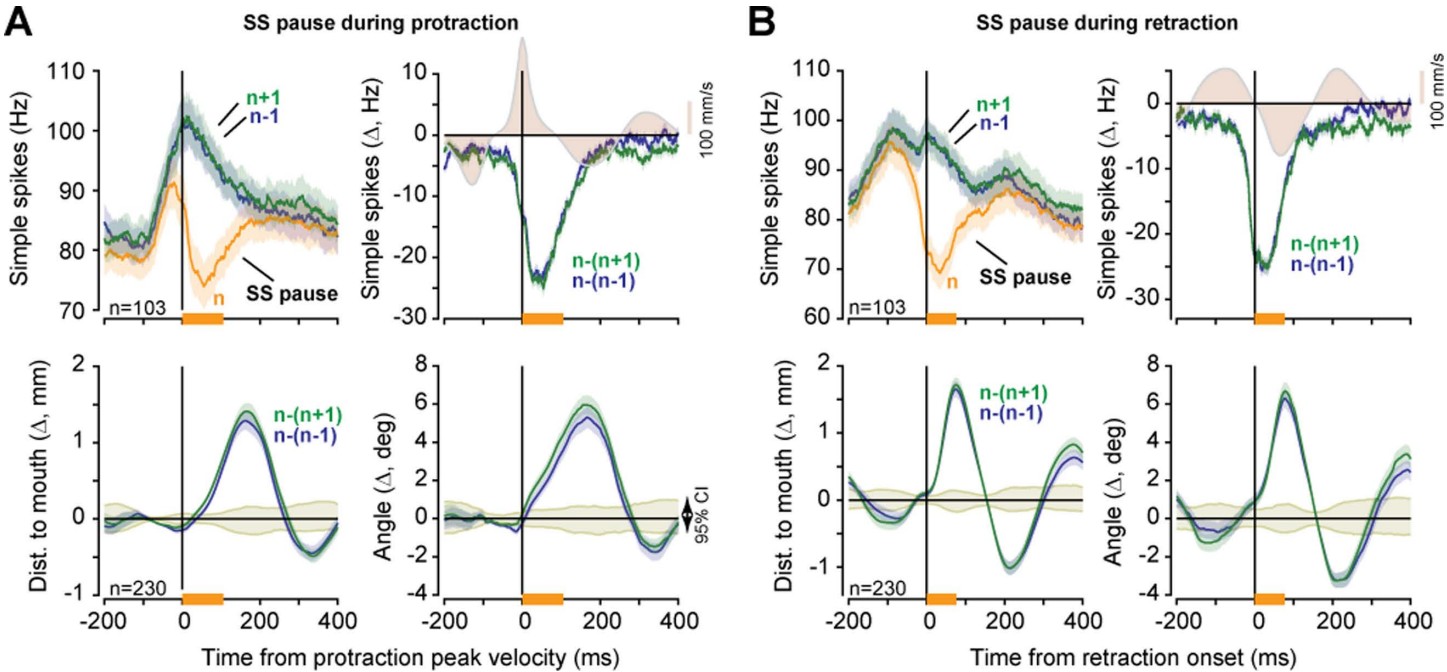

**Fig 6. SS pause without a CS was sufficient to produce hypermetria and bending of the tongue. A.** We selected licks that did not have a CS but nevertheless experienced a long SS pause during the protraction deceleration period. Top row: SS rates for lick n that experienced a pause and licks n-1 and n+1 that did not. Bottom row: change in lick kinematics following the SS pause. **B.** Same as in part A, but for licks that experience a long SS pause during the retraction acceleration period. The data underlying this figure can be found in https://osf.io/wdxu4.

it produced a slowing of the return and also a bending away from the midline (Fig 6B). The trajectory changes were quite similar to what we had observed following a CS-induced SS suppression.

Thus, regardless of whether an SS pause was due to the arrival of a CS or not, the downstream effects were similar: production of forces that extended the tongue, bending it away from the midline.

## The SS rates peaked at deceleration onset, but only if the movement was reward-relevant

Across the cells, the CS rates peaked at approximately the time of maximum protraction velocity (Fig 7A, left column), exhibiting a greater rate for movements toward the ipsilateral side (ipsilateral: 0.21 ± 0.02 Hz, contralateral: 0.16 ± 0.02 Hz, within cell difference, average CS rate, ipsilateral minus contralateral, $t(229) = 28.1$, $p = 2.5E-76$). In contrast, the CS rates declined below baseline before the onset of retraction. Thus, the CS-on phase across the P-cells was protraction.

To analyze the SS activities as a population, we had to consider the fact that during a bout, as the SS rates modulated about a mean, the mean was not stationary (Fig 5A). Rather, the mean SS rates rose or fell before bout onset, then drifted back toward the values before start of the bout, reaching pre-bout levels at the bout ended. Thus, to quantify modulation of SS rates, for each P-cell we considered a 2-s moving window to compute the running average of its firing rate, then computed the SS rates with respect to this mean. The 2-s window was chosen because it was roughly an order of magnitude larger than the duration of a typical lick.

Given that a CS-triggered suppression in SS rates induced downstream forces that pushed the tongue outwards, then if this region of the cerebellum was interested in stopping the ongoing movement, during deceleration of protraction the SS rates should increase, thus commanding forces that would stop the motion of the tongue as it neared the target. Indeed, the SS rates peaked near the onset of lick deceleration, and were larger for contralateral licks (Fig 7A, right column, ipsilateral: $11.58 \pm 1.75$ Hz, contralateral: $12.91 \pm 1.95$ Hz, within cell difference: $t(156) = 13.6$, $p = 2.9E{-}28$).

To test if SS modulation varied with tongue kinematics, we considered two conditions: when licks had the same amplitude but different peak velocities (Fig 7B), and when they had the same peak velocity but different amplitudes (Fig 7C). To consider licks of the same amplitude but different velocity, we quantified lick vigor, defined as the peak speed of the protraction with respect to the speed expected for a lick of the same amplitude [17,31] (S6

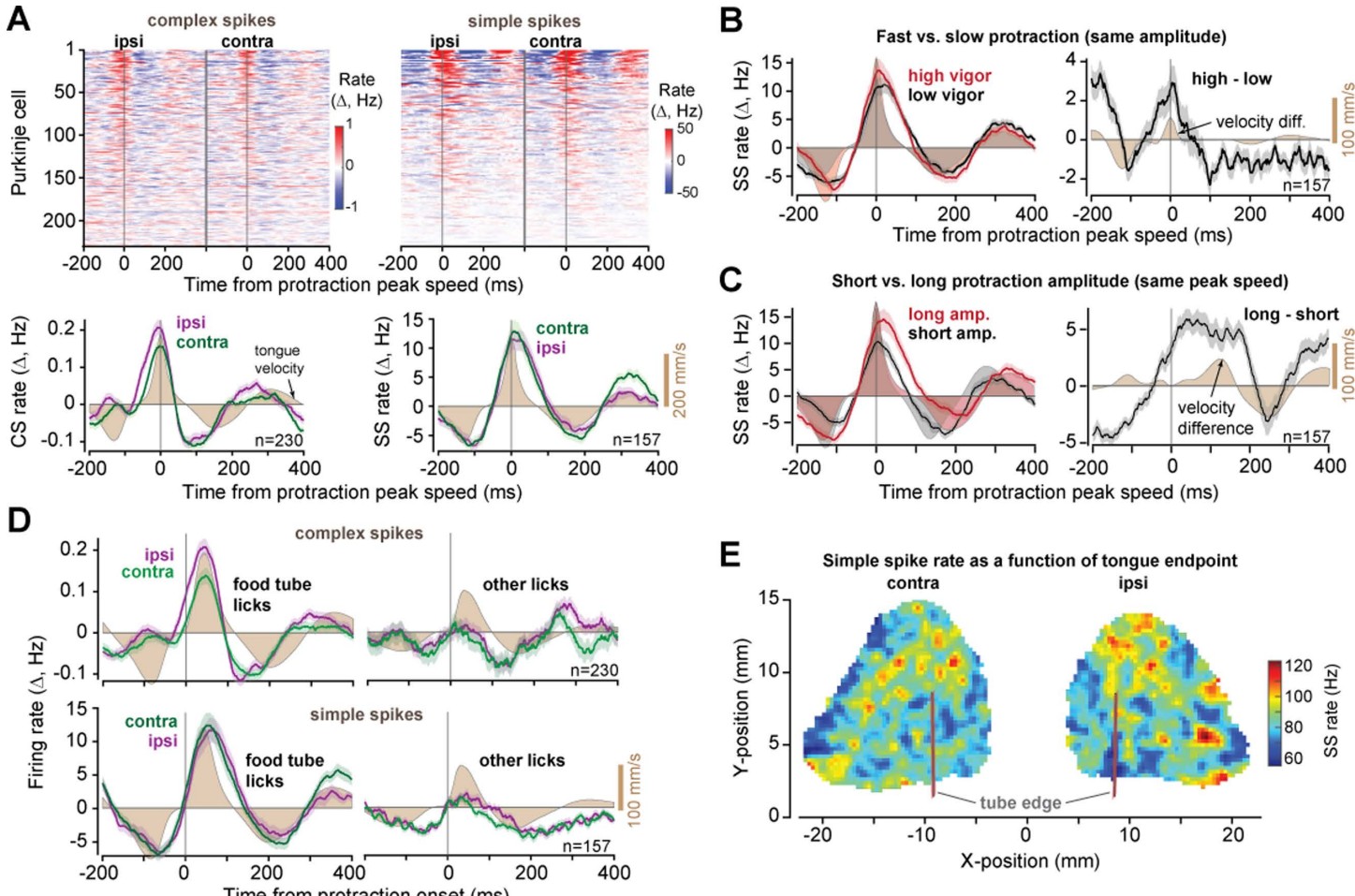

**Fig 7.. SS rates peaked at deceleration onset, but only if the lick was task relevant. A**. CS and SS rates across the population aligned to protraction peek speed. **B**. Left figure shows SS rates for high and low vigor protractions (same amplitude). Right figure shows within cell difference in SS rates for high minus low vigor licks. Shaded regions are tongue speed for high and low vigor licks (left) and change in speed (right). **C**. Left figure shows SS rates for protractions that had long or short amplitudes, but the same peak speed. Right figure shows within cell differences. Shaded regions are tongue speed for long and short licks (left) and change in speed (right). **D**. Modulation of CS and SS rates during task relevant and task irrelevant licks. Light-shaded region is tongue speed for task-relevant movements, while dark shaded region is tongue speed for task irrelevant movements. **E**. SS rates at peak velocity as a function of tongue position at maximum displacement. Error bars are SEM. The data underlying this figure can be found in https://osf.io/wdxu4.

Fig). High vigor licks exhibited greater SS rates at peak protraction speed (Fig 7B, within cell difference, high vigor minus low vigor, 2.92 ± 0.60 Hz, $t(156) = 14.1$, $p = 1.2E{-}29$). That is, licks that required greater deceleration forces accompanied greater SS rates near the onset of deceleration.

Next, we considered licks that had the same peak velocity but different amplitudes (Fig 7C). These two licks began with very similar velocity patterns, reaching on average identical peak velocity, but the SS rates achieved a greater peak rate for the licks that had a longer deceleration period. Moreover, the licks with the longer period of deceleration had a longer period of SS rate increase (Fig 7C, time to baseline crossing, high duration: 126.80 ± 7.68 ms, low duration: 86.02 ± 7.23 ms, paired $t$-test, $t(156) = 7.84$, $p = 6.75E{-}13$). As a result, the SS firing rates at peak velocity tended to increase with the lick's peak amplitude (Fig 7E).

All of these results were for tongue movements in which the subject aimed for the small food tubes. Would the same patterns hold for movements that did not require such precision? To consider this question, we turned to the licks that were not directed toward a food tube, which were often grooming licks. These licks tended to be slower, with a peak speed that was roughly half the speed of the licks directed toward the food tubes (Fig 7D, peak velocity of food tube licks: 290.5 ± 3.6 mm/s, other licks: 153.8 ± 2.5 mm/s). Remarkably, during protraction of these licks the CS rates were an order of magnitude smaller than when the licks were toward the food tubes (Fig 7D, paired $t$-test, tube licks versus grooming, combined directions, $t(229) = -3.27$, $p = 0.00126$). Similarly, the SS rate modulations were much smaller (Fig 7D, right, paired $t$-test, tube versus other, combined directions, $t(156) = 8.6$, $p = 8.44E{-}15$). In sharp contrast, during retraction of these task-irrelevant licks, both the CS and the SS rates were modulated below baseline by amounts that were roughly comparable to the rates of the tube directed licks. That is, the fundamental difference in the P-cell activity between the tube directed and other licks was in the protraction phase, the phase in which control of the tongue required precision.

In summary, when the licks were directed toward the food tube, the CS and SS rates peaked during protraction and were greater when the movement had greater speed. Because SS suppression produced forces that extended the tongue, the fact that SS activity peaked near deceleration onset was consistent with the view that the downstream effects were to decelerate the tongue as it neared its target. Remarkably, both the CS and SS modulation patterns were largely absent when the tongue movements were not aimed at the food tube.

## Error in the tongue's trajectory was reported to the cerebellum via complex spikes

Inserting the tongue into the tube required precision because the opening was only slightly larger than the tongue. As a result, in roughly 15% of the licks the food was inside the tube, but the tongue missed the entrance (S8, S9 and S10 Videos, also S7 Fig). We labeled these as unsuccessful licks because the tongue did not bend enough and instead hit the tube's outer edge. Were these errors reported to the cerebellum?

To visualize the CS patterns as a function of the spatial location of the tongue, we needed to compute $\Pr(CS|x)$, i.e., the probability of producing a CS given that the tongue was at a given location. To arrive at this variable, we began with computing $p(x|CS)$, i.e., the probability density of the position of the tongue's tip $x$, given that a CS occurred at time $t$. To compute this function, we used spike triggered averaging to compute the average position of the tongue during the 50 ms period before the CS. We found that this likelihood depended on whether the tongue successfully entered the tube or not. For licks that were successful and entered the tube, the likelihood $p(x_s|CS)$ was bimodal, exhibiting a peak near lick onset, then a second

peak near the food (Fig 8A, left). For licks that were unsuccessful and did not enter the tube, the likelihood $p(x_u|CS)$ was also bimodal, but now the second peak was around the edge where the tongue had collided with the tube (Fig 8B, left).

We next computed the marginal probability density $p(x)$ for the successful and unsuccessful licks (Fig 8A and 8B, right). This function estimated the probability of the tongue being at a given position during the various licks. We then computed the prior probabilities $\Pr(CS)$, and the ratio of the probabilities $p(x|CS)\Pr(CS)/p(x)$, arriving at the posterior probability $\Pr(CS|x_s)$ and $\Pr(CS|x_u)$. Each posterior estimated the probability of observing a CS as a function of the location of the tongue during successful and unsuccessful licks. Finally, we computed the error-induced spatial pattern of CSs by subtracting the posteriors (Fig 8C). The results revealed a spatial gradient that increased with the tongue's distance from the mouth, suggesting that once we accounted for the CS modulations associated with normal licking, the CS events that remained in the unsuccessful licks tended to occur after the tongue had touched the far end of the tube.

To view these error-specific effects in another way, we plotted the CS rates as a function of time with respect to the tube-touch event. For the successful licks, tube touch occurred when the tongue crossed the tube's edge and entered the tube. In this case, the CS rates following tube touch were depressed (Fig 8D). In contrast, for the unsuccessful licks the tube touch indicated an error, and the CS rates following this event showed a sharp increase (100 ms period following tube touch, CS rate in unsuccessful versus successful licks, paired $t$-test, $t(459) = 8.09$ $p = 5.3 \times 10E{-}15$).

Thus, when the food was inside the tube and the tongue successfully entered it and touched the food, tube touch did not elicit CSs. However, when the food was inside the tube but the tongue failed to enter the tube, now the tube touch event produced CSs.

## Discussion

To quantify how the P-cells in the vermis contributed to control of the tongue, we trained marmosets to make skillful movements, extending and bending their tongue to harvest food

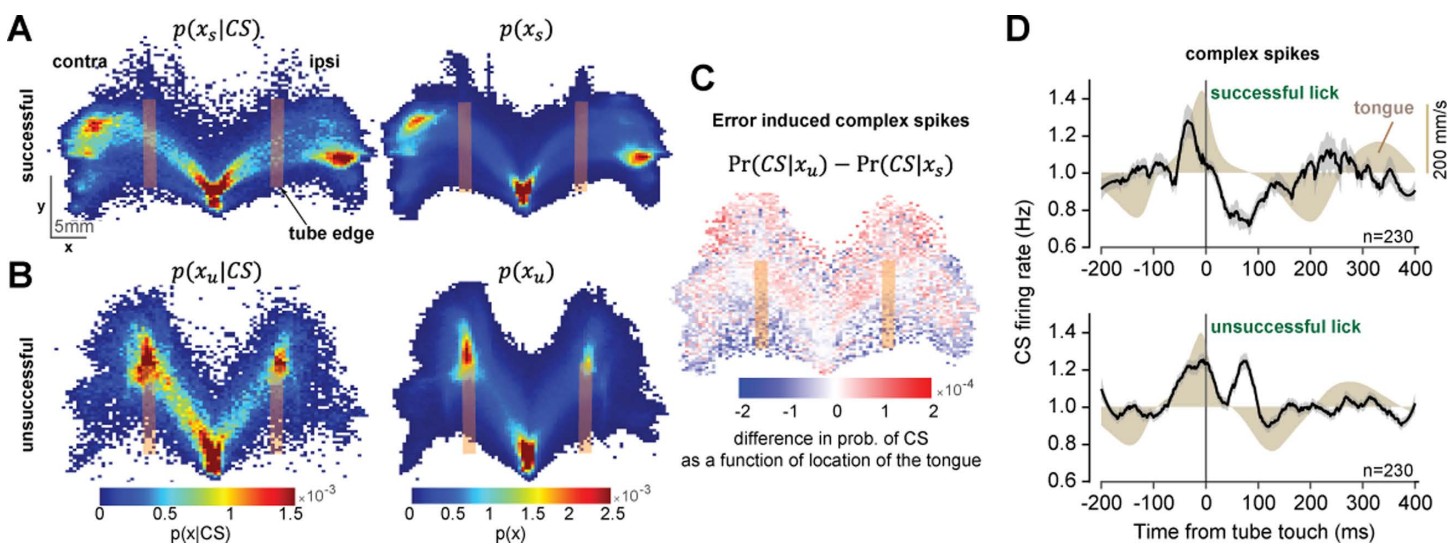

**Fig 8. Error in the tongue's trajectory induced complex spikes. A.** Successful licks: tongue entered the tube and touched the food. Left subfigure shows the spatial likelihood at 25 ms before the CS event. Right subfigure shows the probability of being at a spatial location. **B.** Unsuccessful licks: tongue did not enter the tube and collided with the tube's edge. **C.** Spatial pattern of the error-induced complex spikes (difference between unsuccessful and successful licks in the posterior probability of CS as a function of the tongue's location). **D.** Complex spike rates following tube touch for successful and unsuccessful licks. The data underlying this figure can be found in https://osf.io/wdxu4/.

from small tubes that were placed at 90° with respect to their mouth. Using spike-triggered averaging on the climbing fiber input to each P-cell, we found that if the resulting SS suppression occurred during protraction, there was a disruption in the deceleration phase of the movement, resulting in hypermetria as the tongue approached the target. A suppression that occurred during retraction retarded the tongue's return. When two P-cells were simultaneously suppressed, the kinematic effects magnified. Similar effects were present when there was an SS pause without a preceding CS. Thus, regardless of whether SS production was suppressed during protraction or retraction, the downstream effects were production of forces that pulled the tongue outwards. These results were present for both ipsilateral and contralateral targets, but greater when the target was contralateral.

Because during unperturbed movements the SS rates in the population peaked as the movement began decelerating, we infer that the downstream effects were production of forces in the direction of retraction. This suggests that the contributions of the cerebellum to control tongue movements may be similar to that of the eyes [32,33] and the limbs [7]: steering the movement and stopping it as it nears the target.

## P-cells were modulated only for reward-relevant movements

When the purpose of the movement was to groom the face, during the protraction phase the CS rates remained at baseline while the SS modulations were absent. This observation replicated our observations during saccades: when saccades are aimed toward a reward-relevant target, in lobule VII the CS and SS rates are strongly modulated and the P-cell population predicts when the movement should be stopped [26]. When similar saccades are made without a reward-relevant target, the CS modulations are missing [8] and the SS rates no longer predict deceleration onset [26]. Thus, for both saccades and licking, the cerebellum is engaged only when the movement is reward-relevant.

One reason for engagement of the P-cells during reward-relevant movements may be the greater accuracy requirements of those movements. Target position (for saccades) strongly engages the neurons in the superior colliculus, which appear to transmit that information to the cerebellum via mossy fibers [26]. When the saccade is reward-irrelevant, the encoding of the target location is muted in the colliculus [34–36], as well as the mossy fibers [26]. This implies that when that movement is not reward-relevant, the cerebellum is poorly informed of the goal of the movement. As a result, it cannot assist in predicting when the movement is nearing the target and should be stopped.

Because the colliculus contains a topographic map of tongue movements [37], we conjecture that like saccades, tongue movements that are not reward-relevant will produce muted activity in the colliculus.

## Climbing fibers were most active near movement onset

We were surprised that the CS rates peaked not after the tongue contacted the tube, but around the onset of protraction. This observation reproduced findings of Welsh and colleagues [13] in rats, who recorded from the lateral cerebellum and found that the CS rates peaked near the onset of tongue protrusion, even when the tongue was deafferented. Indeed, in many types of movements, including walking [38], reaching [39–43], and moving the wrist [44], the CS rates peak near movement onset. For example, in the oculomotor vermis, near saccade onset the olivary input informs the P-cells regarding the direction of the upcoming movement [8]. How might the inferior olive be involved in transmitting movement information to the cerebellum?

A key observation is that the inferior olive not only receives input from the superior colliculus [45–47], but that subthreshold stimulation of a region of the colliculus leads to

CS production without producing a movement [48]. Thus, it is possible that the increased CS rates around the onset of protraction reflect activities of regions that initiated the tongue movement, i.e., the motor cortex and the superior colliculus [49]. This prediction remains to be tested with simultaneous recordings from the colliculus, motor cortex, and the cerebellum.

## Climbing fibers reported lick errors

Because the tubes were placed at sharp angles to the mouth, roughly 15% of the licks failed to retrieve the food. In these unsuccessful licks, the tongue did not bend enough and instead collided with the far edge of the tube. The climbing fibers reported this error robustly, exhibiting a strong increase in rates following the touch of the tube. Remarkably, when the touch event was not in error, i.e., the tongue entered the tube, the CS rates were suppressed. Thus, the CS rates signaled a touch that was in error, but not a touch that was expected.

## Using the olivary input to infer a P-cell's contribution to behavior

CSs are rare events, occurring at around once per second. They briefly and completely suppress the SS rates, which induce downstream effects on the cerebellar nuclei [50], potentially producing movements [23]. However, the CS rates are modulated to encode the direction of sensory prediction errors [8,18,51,52]. Thus, it was critical to test that the kinematic effects that we measured following a CS-induced SS suppression were not a consequence of a feedback response to kinematic deviations that occurred before the CS.

We did this by comparing triplets of temporally adjacent licks, finding that while the tongue trajectory preceding the CS event remained within the chance error bounds, the trajectory that followed were robustly different than chance. Notably, the downstream effects of the CS-induced suppression were in the same direction, i.e., extension of the tongue, regardless of whether the CS events occurred during protraction or retraction. This is notable because the CS rates were maximum during protraction, and minimum during retraction, yet their downstream effects were the same: pull the tongue outwards.

The idea that the olivary input may affect ongoing movements was noted by Ebner and colleagues during reaching movements [24], and subsequently observed during saccades [25]. For example, during saccades following the CS-induced SS suppression, the eyes are pulled in the CS-on direction of the P-cell. This is consistent with the fact that optogenetic increase in the SS rates suppresses the cerebellar nuclei [53], pulling the eyes approximately in direction CS+180 [54]. While relying on the stochasticity in the olivary input has the disadvantage of lacking a causal manipulation, it has the advantage of relating the kinematic effects to CS-on properties of each P-cell, something that would not be possible with large-scale optogenetic stimulation.

## A long SS pause had the same effect on behavior as a CS-induced SS suppression

A CS is followed by SS suppression, but pauses in SS production can also occur because of other reasons [28], including inhibitory input from the molecular layer interneurons [55]. Suppression of SS production can result in an increase in the activity of nucleus neurons, resulting in movement of the limb [56]. Here, we found that the effects of CS-induced SS suppression on tongue kinematics were largely the same as SS pauses that were not due to the arrival of a CS. In both cases, the result was a force that pulled the tongue outwards and bent it away from the midline. This implies that the downstream effects on kinematics were not due to the arrival of an input from the inferior olive, but rather the SS pause in the parent P-cell.

## A cortico-cerebellar network for control of the tongue

As our marmosets prepared to initiate a licking bout, the P-cells exhibited a ramping activity. Previous reports have noted that during this period there is ramping activity in the tongue regions of the motor cortex [57], as well as in the fastigial [57] and the dentate nuclei [58]. Inhibiting the tongue-jaw region of the motor cortex in mice disrupts licking to the sides, and prevents the onset and the termination of lick sequences [59]. Inhibiting the fastigial during the ramping period disrupts planning of the movement and removes the direction selectivity that the motor cortical cells exhibit [57], while inhibiting the dentate disrupts the ramping activity in the motor cortex [58]. Similarly, exciting the P-cells in the vermis abolishes the ramping activity in motor cortex during the delay period of a decision-making task [60]. This implies that as one prepares to initiate a movement, the rising activity in the motor cortex is controlled via a loop through the cerebellum.

Our results here suggest that once the tongue movement begins, there is a specific role for the cerebellum in producing forces that would stop the protraction, especially if that movement is reward-relevant. We speculate that the cerebellum is informed via the mossy fibers of two kinds of information: the location of the target, and a copy of the ongoing motor commands [26]. The function of this region of the cerebellum may be to use the copy of the motor commands to predict when the tongue is about to reach the target and aid in production of commands that would stop the outward movement.

## Medial and lateral parts of the cerebellum may contribute to different aspects of tongue movements

In humans, the tongue region of the cerebellum extends from lobule VI in the vermis laterally to the hemispheres [61–63]. Dysarthria is principally associated with damage in the paravermal regions of the cerebellum [64]. In macaques, stimulation of the fastigial nucleus moves the tongue in the protraction-retraction axis, while stimulation of the dentate nucleus moves it in the lateral-medial axis [10]. In mice, activation of the P-cells in the lateral regions of lobule VI and VII during protraction bends the tongue toward the ipsilateral side [65]. When we consider these results together with our observations here, what emerges is the conjecture that the P-cells in the vermis are important for control of protraction/retraction, but the P-cells in the paravermis and hemispheres have a different role, possibly in controlling how the tongue bends.

## Toward a general model of cerebellar control of movements

Like the SS rates, the CS rates peaked near protraction peak velocity, then fell below baseline before the onset of retraction. Thus, for both ipsilateral and contralateral movements, the "CS-on action" across the P-cells was protraction, while the "CS-off action" was retraction. Notably, the downstream effects of SS suppression were to extend the tongue. As a result, there was a correspondence between the vector that described the CS-on action, and the vector that described the effects of SS suppression. This fact is notable because the same principle holds for P-cells in the oculomotor region of the cerebellum during saccadic eye movements [25,26]: the olivary input to an oculomotor P-cell is most active when a saccade is planned in direction CS-on, and the downstream effects of that P-cell's SS suppression is to pull the eyes also in direction CS-on. Thus, for both eye and tongue movements, the olivary input provides a vector-based coordinate system [26] with which one might estimate the downstream contributions of a P-cell to control the movement [66,67].

The key theoretical idea is that the inferior olive organizes the cerebellum so that the P-cells are placed in competition with each other: for every P-cell that has a particular CS-on,

effecting movements along a particular potent vector, there is another that prefers the opposite vector [26]. Unfortunately, here we could not apply this theory to organize the P-cells into antagonist populations because nearly all the cells in our database had a CS response that peaked during protraction. However, our theory [26] predicts that there should be P-cells whose climbing fiber inputs prefer retraction. In these P-cells, the SS suppression should pull the tongue inward. If these P-cells exist, then their SS pattern would be antagonistic to the SS pattern of the P-cells we found here, resulting in a population response in which P-cells would compete with each other, perhaps producing a sum of activity that is a burst-pause pattern, inhibiting then disinhibiting the nucleus as the tongue approaches the target.

Lingual dysfunction accompanies a host of symptoms, including vocal muscle dystonia [68], problems in swallowing [69], and dysarthria [2,64,70], all of which share a link to the cerebellum. Rehabilitation or cures for these symptoms will require a much better understanding of how the cerebellum contributes to the control and learning of tongue movements. Marmosets are exceptionally skilled in shaping and twisting their tongue, using it almost like a finger. This makes them an attractive new model to study the neural control of a body part that is essential for our existence.

## Methods

Data were collected from three marmosets, *Callithrix Jacchus*, 2 male and 1 female, 350–370 g, subjects 125D (Mirza), 59D (Ramon), and 132F (Charlie), during a 3.5-year period. The marmosets were born and raised in a colony that Prof. Xiaoqin Wang has maintained at the Johns Hopkins School of Medicine since 1996. The procedures on the marmosets were approved by the Johns Hopkins University Animal Care and Use Committee in compliance with the guidelines of the United States National Institutes of Health (protocol number PR22M285).

### Data acquisition

Following recovery from head-post implantation surgery, the animals were trained to make saccades to visual targets and rewarded with a mixture of applesauce and lab diet [16]. Visual targets were presented on an LCD screen. Binocular eye movements were tracked at 1,000 Hz using EyeLink in subjects R and M, and 2,000 Hz using VPIX in subject C. Tongue movements were tracked with a 522 frame/s Sony IMX287 FLIR camera, with frames captured at 100 Hz.

We performed MRI and CT imaging on each animal and used the imaging data to design an alignment system that defined trajectories from the burr hole to various locations in the cerebellar vermis [16], including points in lobules VI and VII. We used a piezoelectric, high-precision microdrive (0.5 micrometer resolution) with an integrated absolute encoder (M3-LA-3.4-15 Linear smart stage, New Scale Technologies) to advance the electrode.

We recorded from lobules VI and VII of the cerebellum (Fig 1C) using quartz insulated 4 fiber (tetrode) or 7 fiber (heptode) metal core (platinum/tungsten 95/05) electrodes (Thomas Recording), and 64 channel checkerboard or linear high-density silicon probes (M1 and M2 probes, Cambridge Neurotech). We connected each electrode to a 32- or 64-channel head stage amplifier and digitizer (RHD2132 and RHD2164, Intan Technologies, USA), and then connected the head stage to a communication system (RHD2000 Evaluation Board, Intan Technologies, USA). Data were sampled at 30 kHz and band-pass filtered (2.5–7.6 kHz).

The silicon probes arrived with a polymer coating on the contacts that degraded with each insertion into the brain [71]. This degradation increased the impedance of the electrodes and dramatically reduced the ability of the probe to isolate neurons. We found it essential to rejuvenate the silicon probes by stripping and then re-depositing the polymer coating after every 3–4 insertions into the brain [71].

## Behavioral protocol

Each trial began with fixation of a center target after which a primary target appeared at one of eight randomly selected directions at a distance of 5°–6.5°. As the subject made a saccade to this primary target, that target was erased, and a secondary target was presented at a distance of 2°–2.5°, also at one of eight randomly selected directions. The subject was rewarded if following the primary saccade, it made a corrective saccade to the secondary target, landed within 1.5° radius of the target center, and maintained fixation for at least 200 ms. The food was provided via two small tubes (4.4 mm diameter), one to the left and the other to the right of the animal, positioned at 90° with respect to the mouth. A successful trial produced a food increment in one of the tubes and would continue to do so for 50–300 consecutive trials, then switch to the other tube. Because the food increment was small, the subjects naturally chose to work for a few consecutive trials, tracking the visual targets and allowing the food to accumulate, then stopped tracking and harvested the food via a licking bout. The subjects did not work while harvesting, and often fixated the tube. As a result, the behavior consisted of a work period of targeted saccades, followed by a harvest period of targeted licking, repeated hundreds of times per session.

We measured eye movements during all phases of the task, including the bouts of licking. The monkeys tended to fixate the tube while licking. We analyzed tongue movements using DeepLabCut [29]. Our network was trained on 89 video recordings of each subject with 15–25 frames extracted and labeled from each recording. The network was built on the ResNet-152 pre-trained model, and then trained over $1.03 \times 106$ iterations with a batch size of 8, using a GeForce GTX 1080Ti graphics processing unit. A Kalman filter was further applied to improve quality and smoothness of the tracking, and the output was analyzed in MATLAB to quantify lick events and kinematics. We tracked the tongue tip and the edge of the food in the tube, along with control locations (nose position and tube edges). We tracked all licks, regardless of whether they were aimed toward a tube, or not. Food-tube licks were further differentiated based on whether they aimed to enter the tube (inner-tube licks) or hit the outer edge of the tube (outer-edge licks). If any of these licks successfully contacted the food, we labeled that lick as a success (otherwise, an unsuccessful lick).

## Tracking the tongue

The following videos provide examples of the various types of licks, along with the kinematic measures that we used to track each movement: S1–S10 Videos. Licks were categorized based on heuristics that considered the position of the tongue relative to the tube opening and the food. We tracked four regions of the tongue consisting of the tip, the midpoint, and the left and right edges. The midpoint was computed based on the distance between the tip marker and the opening of the mouth, while the left and right edges were computed based on the mid-distance between the tip and midpoint, positioned at max laterality. Furthermore, we tracked the left and right edges of the opening of each reward tube as well as the densest edge of the food contained within.

Licks were labeled as reward-seeking when the region of the tongue within the marker overlapped with the edge of the tube coordinates. Alternatively, licks were labeled as grooming when no overlap occurred. Reward-seeking licks were further labeled into subcategories, consisting of inner- and outer-tube licks. Inner-tube labels were assigned when the tip, left, and right tongue markers remained within the bounds of the tube edge markers. Outer-tube labels were assigned when at least one marker exited the tube boundaries, conditioned on the tip having remained within at least 5 mm of the tube opening.

Additional labels were assigned to each reward-seeking licks, categorizing them as either successful or unsuccessful licks. In all cases, overlap with food dictated these labels. Thus,

to call a given lick an unsuccessful lick, the position of the food within the tube, relative to the tongue, was considered. For example, consider a scenario in which the food is depleted, requiring an inner-tube lick to scoop out the remaining bolus. If the lick entered the tube and thus touched the food, it was considered a success. If it did not enter the tube and thus did not touch the food, it was considered an unsuccessful lick.

## Neurophysiological analysis

We used OpenEphys [72] for electrophysiology data acquisition, and then used P-sort [30] to identify the SSs and CSs in the heptodes and tetrodes recordings. We used Kilosort and Phi [73] to identify the spikes for the silicon probes. SSs and CSs instantaneous firing rate were calculated from peri-event time histograms with 1 ms bin size. We used a Savitzky–Golay filter (2nd order, 31 datapoints) to smooth the traces for visualization purposes.

Many P-cells in lobules VI and VII of the vermis were modulated during licking as well as during saccades. Our data here were selected from recordings that isolated P-cells with strong tongue-related activity. The strength of behavioral modulation for each P-cell during saccades and licks was quantified using a $z$-score (S2B Fig). This $z$-score was calculated for each behavior via the range of the P-cell's average stimulus-aligned response divided by the standard deviation of this range, as computed across 2,000 permuted responses. Range was defined as the maximum change in firing rate from pre-behavior to post-behavior for a given response. This approach relies on the notion that if a cell is responsive to a given stimulus, it will exhibit both strong response (high range) and a consistent response (low standard deviation of range values). Consistent with earlier work [9], the threshold for significant modulation during licking was set at a $z$-score of 3.

CS baseline firing rates were computed by dividing the total number of spikes by the duration of the entire recording. SS baseline firing rates were computed using two different methods depending on the analysis. For bout-related responses, baseline was defined as the average firing rate in a 300 ms window preceding bout onset by 700 ms, i.e., during the $[-1,000$ to $-700]$ ms period. However, to analyze the activities during individual licks, because the rates were not stationary but gradually changing from the first to the last lick in the bout, baseline SS rates were computed using the average firing within a sliding window of 2 s, consisting of 5–6 licks.

To explore how the SS rates changed with the kinematic parameters of the lingual movements, we visualized the firing rates as a function of tongue endpoint position. The firing rates of each P-cell during maximal tongue velocity were computed on a trial-by-trial basis and associated with the spatial coordinates corresponding to the endpoint of that trial's lick. Single-trial spike data was smoothed with a Savitzky–Golay filter. Spatial coordinates were standardized across animals such that all contralateral licks appear to the left and all ipsilateral licks to the right. A 100×50 grid was mapped onto the full range of tongue endpoint values, and the population firing rates at each point were estimated using a natural neighbor interpolation, effectively weighing contributions of neighboring firing rate values based on proximity. The interpolated surface was then smoothed with a 2-D Gaussian filter to produce a continuous heatmap. To ensure population-level robustness of firing rate values, a cell coverage mask was then applied over the heatmap, removing any grid points that did not have at least 75% of the available PCs (118/157 SS cells).

## Computing the kinematic effects of CS-induced SS suppression

For each P-cell we considered triplets of tube-directed licks $\{n-1, n, n+1\}$, where all three licks were of the same type, i.e., contacted the same part of the tube (edge or inner). We

then selected the subset of triplets in which there was a CS at only a single period in lick $n$, but no CS during any period in the two neighboring licks $n-1$, and $n+1$. We then compared tongue trajectories between the lick that had a CS with the two neighboring licks, i.e., $n-(n-1)$ and $n-(n+1)$.

## Computing the kinematic effects of SS pauses

To assess if the perturbation of tongue movements was a consequence unique to the presence of a CS, or rather the suppression of SSs, we considered the effect of SS pauses on the tongue trajectory during licking, i.e., long ISI events that were not preceded by a CS.

For each P-cell, we selected the subset of all licks of the same type towards the same direction in which no CSs occurred at any point in the movement. Let us call these the NoCS licks. Working only with the NoCS licks, we sought to identify licks in which during a phase of interest (e.g., protraction deceleration), there was a long pause in the SS production. However, we had to ensure that if there was a long pause in one phase of the lick, it did not also occur in other phases of the same lick. That is, like the CS analysis, to be eligible for this analysis a long SS pause had to occur only once during the lick.

There were four phases for each lick (protraction acceleration and deceleration, retraction acceleration and deceleration), i.e., $p = 1, \ldots, 4$. For each lick $n$, during each phase $p$, we found the duration of the longest ISI that originated in that phase (regardless of whether it extended into the next phase) and labeled it as $t_n^{(p)}$. Next, for each phase, we found the distribution of $t_n^{(p)}$. Licks with zero SSs during the given phase were excluded from this distribution.

For example, suppose we were interested in labeling the licks in which during phase 1 there was a long pause. A lick with a long pause in phase 1 could not have also had a long pause in another phase of that same lick. We found the distribution of $t_n^{(2)}$, the distribution of $t_n^{(3)}$, and the distribution of $t_n^{(4)}$, and then for each phase selected the top quartile (25% longest ISIs). We removed the licks with a long pause in phase 2–4 for consideration. From among the remaining licks, we formed the distribution of $t_n^{(1)}$, found the top quartile, and labeled those as having a long pause during phase 1. We labeled the remaining 75% of licks in this population as not having a long pause during this phase.

We selected the subset of triplets in which lick $n$ had a long SS pause in only one phase of the movement, but no SS pause occurred in any phase of the two neighboring licks. We then compared tongue trajectories between the lick that had a pause with the two neighboring licks. Traces were averaged within directions and then across directions for each cell.

## Computing the effects of trajectory error on climbing fiber activity

Roughly 15% of the licks failed to enter the tube and did not touch the food (S5 Fig). To visualize the CS patterns as a function of the spatial location of the tongue, we began with computing $p(x(t-25)|CS)$, i.e., given that a CS occurred at time $t$, the likelihood of the tongue's tip location $x$ at time $t-25$ ms. We did this by averaging the position of the tip of the tongue during the 50 ms period before the CS event. We separated the licks into successful licks (tongue entered the tube and touched the food) and unsuccessful licks (tongue touched the tube but neither entered it nor touched the food). The result was the likelihood $p(x_s|CS)$ for the successful licks and $p(x_u|CS)$ for the unsuccessful licks.

We next computed the marginal probability density $p(x)$ for each lick type, the prior $\Pr(CS)$ (from the average CS rate during a lick of that type, using a 50 ms time bin), and then the ratio of the probabilities $p(x|CS)\Pr(CS)/p(x)$. Finally, we computed the error-induced spatial pattern of CSs by subtracting this ratio for the successful licks from the same ratio for the unsuccessful licks. To reduce the noise associated with the far edges of each probability

density function, for each ratio we considered values that were in the 95% quantile of its distribution.

### Statistical analysis

In order to compare the measured effect of SS suppression on tongue trajectory with what would be expected to happen simply due to chance, we computed the bounds for the null hypothesis. To do so, we used bootstrapping to compute 95% CIs. We shuffled the assignment of CS tags from the lick in which it had occurred to a randomly assigned lick of the same type. Using this pseudo-data, we then selected triplets of consecutive tube-directed licks and computed trajectory differences among neighboring licks, averaging $n-(n-1)$ and $n-(n+1)$. We computed this expected value for each cell, computed a mean across all the cells, and then repeated the shuffling 30 times to compute 95% CIs.

To plot the data in the various figures, which are available here, we used the following publicly available MATLAB packages: suplabel.m, boundedline.m, and violinplot.m.

## Supporting information

**S1 Video. A sequence of five licks to the right tube.** The top two plots show trajectory of the tip of the tongue and a geometric representation of four markers on the tongue. The second row shows the displacement, velocity, and angle of the tongue as a function of time. The third row shows the distance of the tip of the tongue to the food in the left and the right tube. The first 3 licks are successful and enter the tube and contact the food. In the fourth and fifth licks, the tongue fails to enter the tube and does not contact the food. These licks are unsuccessful and are analyzed in Fig 8.
(GIF)

**S2 Video. Example of a grooming lick.** These licks aim to clean the regions around the mouth and are not aimed toward the food tubes.
(GIF)

**S3 Video. Example of a grooming lick.**
(GIF)

**S4 Video. Example of a bout of grooming licks.**
(GIF)

**S5 Video. Example of an outer-tube lick.** The food has accumulated beyond the edge of the tube and the subject begins the bout by licking the food near the edge.
(GIF)

**S6 Video. Example of an inner-tube lick.** The food is deep inside the tube and the subject enters the tube and scoops the food out.
(GIF)

**S7 Video. Example of an inner-tube lick.**
(GIF)

**S8 Video. Unsuccessful lick.** The food is inside the tube, but the lick fails to enter it and instead goes under the tube.
(GIF)

**S9 Video. Unsuccessful lick.** The food is inside the tube, but the lick fails to enter it and instead goes to the outer edge.
(GIF)

**S10 Video. Unsuccessful lick.** The food is inside the tube, but the lick fails to enter it and instead collides with the edge.
(GIF)

**S1 Fig. The kinematic properties of licks in each monkey.** The data underlying this figure can be found in https://osf.io/wdxu4/files/osfstorage.
(AI)

**S2 Fig. Properties of the P-cells in the database. A.** The number of neurons recorded from each marmoset. CS only refers to P-cells for which only the complex spikes were recorded. CS & SS refers to P-cells for which both the CS and SS were recorded. SS only refers to putative P-cells for which only the SS were recorded. **B.** Modulation index of each P-cell during licking and saccades. The values indicate the mean ± SD of each distribution. **C.** Simple spike modulation in two example P-cells. The activities in each cell are aligned to lick bout onset, and saccade onset, both for reward-relevant movements. The data underlying this figure can be found in https://osf.io/wdxu4/files/osfstorage.
(AI)

**S3 Fig. CS-induced SS suppression did not affect movements during protraction acceleration period. A.** Suppression took place during the acceleration period of protraction. Traces show average tongue trajectory during the acceleration period of protraction for each P-cell during suppressed and control licks. Ipsilateral licks are shown to the left and contralateral to the right. Heatmap quantifies change in endpoint trajectory between suppressed and control licks for each cell. Period of suppression is indicated by the orange bar at the bottom of the heatmap. **B**. Top row: SS rates for licks $\{n-1, n, n+1\}$, where only lick $n$ experienced a CS. Filled color curves indicate tongue velocity. Second row: trajectory of the tongue in lick $n$ as compared to its two temporally neighboring licks. Trajectory is measured via distance from tip of the tongue to the mouth and angle of the tip with respect to midline. The filled region is 95%CI. **C**. Distance to mouth and angle in lick $n$ as compared to neighboring licks. Shaded region is 95%CI. The data underlying this figure can be found in https://osf.io/wdxu4/files/osfstorage
(AI)

**S4 Fig. Kinematic effects of CS-induced SS suppression in Cartesian coordinates.** Left plot shows the change in the position of the tip of the tongue at the end of protraction in movements that experienced a CS during the deceleration period of protraction, with respect to movements that did not experience a CS during any period. Right plot shows the change in position of the tip of the tongue as measured at peak retraction velocity, comparing movements that experienced a CS during the retraction acceleration period with movements that did not experience a CS in any period. Each line is the average effect for a single P-cell. P-cell suppression during protraction produced hypermetria, whereas the suppression during retraction resulted in slowing. The data underlying this figure can be found in https://osf.io/wdxu4/files/osfstorage
(AI)

**S5 Fig. There was little or no change in lick kinematics when the CS-induced SS suppression occurred before lick onset.** We chose the period following the completion of one lick and the start of the next lick, which was on average 30 ms in duration, i.e., the period in which the tongue was in the mouth. Top row: SS rates, aligned to lick onset, protraction peak speed, and retraction onset. Second row: change in SS rates. Third row: change in tongue displacement. Fourth row: change in tongue angle. The filled region is 95% confidence interval. Error

bars are SEM. The data underlying this figure can be found in https://osf.io/wdxu4/files/osfstorage

(AI)

**S6 Fig. Peak velocity as a function of lick amplitude during protraction and retraction for task relevant and task irrelevant licks in each subject.** Error bars are SEM.

(AI)

**S7 Fig. Percent of licks in which the food was inside the tube, but the tongue missed the tube's entrance and did not contact the food.** The data underlying this figure can be found in https://osf.io/wdxu4/files/osfstorage

(AI)

**S1 Table. Distribution of tongue- and eye-modulated P-cells in the vermis region of the cerebellum.**

(DOCX)

## Author contributions

**Conceptualization:** Reza Shadmehr.

**Data curation:** Mohammad Amin Fakharian, Alden M. Shoup, Jay S. Pi, Ehsan Sedaghat-Nejad, Simon P. Orozco, In Kyu Jang, Vivian Looi, Toren Arginteanu.

**Formal analysis:** Paul Hage, Hisham Y. Elseweifi, Nazanin Mohammadrezaei, Alexander N. Vasserman, Reza Shadmehr.

**Investigation:** Paul Hage.

**Methodology:** Paul Hage.

**Software:** Mohammad Amin Fakharian.

**Validation:** Paul Hage, In Kyu Jang, Vivian Looi, Toren Arginteanu.

**Visualization:** Paul Hage, Hisham Y. Elseweifi, Nazanin Mohammadrezaei, Alexander N. Vasserman.

**Writing – original draft:** Reza Shadmehr.

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
