## [Editor Report · Decision Letter 0]

12 Aug 2024

Dear Dr Shadmehr, 

Thank you for submitting your manuscript entitled "Control of tongue movements by the Purkinje cells of the cerebellum" for consideration as a Research Article by PLOS Biology.

Your manuscript has now been evaluated by the PLOS Biology editorial staff as well as by an academic editor with relevant expertise and I am writing to let you know that we would like to send your submission out for external peer review.

Once your full submission is complete, your paper will undergo a series of checks in preparation for peer review. After your manuscript has passed the checks it will be sent out for review. To provide the metadata for your submission, please Login to Editorial Manager (https://www.editorialmanager.com/pbiology) within two working days, i.e. by Aug 14 2024 11:59PM.

Kind regards,

Suzanne de Bruijn, PhD

Associate Editor, PLOS Bioloygy

On behalf of,

Christian

Christian Schnell, PhD, 

Senior Editor

PLOS Biology

cschnell@plos.org

---

## [Decision Letter · Decision Letter 1]

17 Oct 2024

Dear Dr Shadmehr,

Thank you for your patience while your manuscript "Control of tongue movements by the Purkinje cells of the cerebellum" was peer-reviewed at PLOS Biology. Please allow me to apologize again for the long delay in sending our decision. Unfortunately, the previous Academic Editor is currently unavailable and it was a bit tricky to find a new one while I was traveling in the past couple of weeks. In any case, your manuscript has now been evaluated by the PLOS Biology editors, an Academic Editor with relevant expertise, and by several independent reviewers. 

In light of the reviews, which you will find at the end of this email, we would like to invite you to revise the work to thoroughly address the reviewers' reports.

As you will see below, the reviewers are quite positive about your study, but they note a few points where additional analyses and textual clarifications would improve the manuscript. 

Given the extent of revision needed, we cannot make a decision about publication until we have seen the revised manuscript and your response to the reviewers' comments. Your revised manuscript is likely to be sent for further evaluation by all or a subset of the reviewers.

**IMPORTANT - SUBMITTING YOUR REVISION**

*Re-submission Checklist*

*Published Peer Review*

*PLOS Data Policy*

*Blot and Gel Data Policy*

Sincerely,

Christian

Christian Schnell, PhD

Senior Editor

PLOS Biology

cschnell@plos.org

REVIEWS:

Reviewer #1 (Detlef H. Heck): The manuscript by Hage et al. describes interesting findings from well-designed experiments to determine the influence of complex spike-induced changes in Purkinje cell simple spike activity on goal directed tongue movements in mamosets. The strength of the approach chosen is that recordings are performed in a primate and the authors did not manipulate large populations of Purkinje cells with population synchronizing optical, chemical or electrical stimulations but instead recorded naturally occurring instances of complex spikes during a large number of licks performed to gain food rewards. The results are in surprising in the sense that a complex spike observed in just a single Purkinje cell (causing the expected brief pause in simple spike activity) had a significant effect on tongue protrusion and retraction if it occurred during certain phases of the tongue movements, and that this effect was significantly increased when a complex spike occurred simultaneously in two observed Purkinje cells. The overall direction of the effect on tongue movements indicates that simple spike activity serves to slow down tongue movements to properly stop at the desired target, which is a finding that resonates with cerebellar involvement in both eye or limb movements. 

There are no major concerns with this manuscript. The experimental design and data analysis see solid, and the manuscript is well written with some minor issues.

Minor concerns:

The figures are not cited in order throughout the text. The authors should rearrange the figures to fix this issue.

The recording location is mentioned early in the manuscript but should also be mentioned in the method section, where it just says "cerebellum".

Complex overlapping bar graphs (e.g. Fig. 2 C) are sometimes difficult to read. Consider making them larger.

The tongue trajectory plots in Fig. 3 A and G are difficult to interpret. Maybe enlarge them or describe better what they are showing.

Legend for Fig. 3A refers to an "orange bar" that this reviewer cannot find in the figure.

Reviewer #2: In this manuscript, Hage et al. present their research on cerebellar control of tongue movements in marmosets. The study focuses on how vermal Purkinje cells (P-cells) contribute to the precision of tongue movements during food harvesting. Using electrophysiological recordings and high-speed video tracking, the authors analyzed the relationship between complex spikes (CS), simple spikes (SS), and tongue kinematics. They found that CS-induced pauses in SS firing led to hypermetria during protraction and slowing during retraction of the tongue. The hypermetria increased with synchronous CS firing across multiple P-cells, demonstrating the importance of synchronous CS firing. Surprisingly, this phenomenon was consistent across the recorded P-cells despite heterogeneity in their SS firing patterns. Finally, they explored the encoding of kinematic parameters by SS rate, which showed significant modulation by peak velocity and duration of deceleration. Notably, this modulation became much less significant when the tongue movement was not directed toward an external goal (food) but was used for grooming.

This is an interesting study with many fascinating findings based on a novel paradigm. However, I found several confusing points that need clarification. Also, the authors need to improve their presentation of the data, which can help with clarification.

Major issues:

1. P-cell selectivity to tongue movement

The data is obtained from the vermis including lobule VI/VII during animals performing the saccade task. An important question here is whether the P-cells in this dataset were eye movement-sensitive, and, if so, how many of them are in the dataset. Regardless of whether the answer is yes or no, this would be an interesting finding to report.

2. SS pauses

This study focuses on pauses in SS firing caused by CSs. CSs induced short pauses of ~15 ms while they are associated with significant differences in tongue movements. However, P-cells can pause SS firing without CSs, contributing to multiplexing different types of information into the P-cell SS output (Hong et al., eLife, 2016; Brown et al., Neuron, 2024). Furthermore, there is evidence that those SS pauses synchronously begin or terminate across multiple P-cells (Jaeger, J Comp Neurosci, 2003; Shin and De Schutter, J Neurophys, 2006; Hong et al., eLife, 2016), similar to the CS-induced pauses in this study. Therefore, it is necessary to verify whether the changes in the tongue movements are specific to the CS-driven pauses or general SS pauses.

Specifically, I suggest analyzing how the distribution of pause lengths after CSs (~15ms) compares to that of all SS interspike intervals (ISIs) to verify if there are large ISIs that interrupt fast simple spiking, or SS pauses. If so, the authors should test if non CS-driven pauses can evoke changes in the tongue movements. Regardless of what the authors find from this analysis, it would be valuable information.

3. Data presentation

I had some difficulty understanding the left and right panels in Fig. 1D showing different segments in the recording data. It would be really helpful if the authors put an extended trace of the data showing how those panels are located on a longer time scale, similar to Fig 5A, which greatly aided my understanding of what the data look like.

Additionally, Fig 5A and other figures seem to suggest that P-cell activity is quasi-periodic as the animal repeats similar tongue movements. Was there any reason not to use a cyclic description (e.g., amplitude & phase) of tongue movement for analysis? It would be beneficial if the authors could comment on whether the tongue movements were sufficiently cyclic or not.

4. Encoding of kinematic parameters by SSs

The authors' explanation about the longer deceleration case from lines 304 to 308 is quite confusing. If the motion starts with a certain initial speed but takes longer to decelerate until stopping, one would expect the magnitude of deceleration and therefore the force (and consequently the SS rate according to the authors' interpretation) to be smaller. Additionally, the travel distance would be longer in this case, which seems consistent with the increased distance to the mouth observed with the SS pause (rate decrease). Maybe I am misreading this passage, but it certainly needs clarification to avoid misinterpretation.

Also, tt would be valuable if the authors could develop a parametric model of the SS firing instead of comparing two bins. In most previous studies, the SS rate linearly encoded kinematic parameters. Whether the tongue movement follows the same pattern or not would be a very interesting finding that could provide deeper insights into how the cerebellum controls tongue movements.

Minor issues:

1. In Fig. 1B and D, small numbers (possibly data IDs or version numbers) should be removed.

2. The authors state neurons were recorded in "various locations" in the vermis including lobule VI and VII. Can they clarify this by describing how the recording locations are distributed? How many neurons were confirmed to be in lobule VI/VII? A more precise breakdown of recording locations would be helpful.

Reviewer #3 (Jesse H Goldberg): This paper examines Purkinje cell discharge patterns in a dexterous lick task designed for marmosets. 

Major strengths:

1) Paper is very well written.

2) The behavioral task and choice of model system to examine tongue control is innovative and well executed/justified. Natural tongue dexterity is leveraged in this well designed task.

3) The authors did rigorous mapping to identify the part of the vermis where they seemed to be 'swimming' in lick related signals. This is important and difficult work that significantly advances the field and sets the stage for future work. 

4) The authors identified consistent patterns of Prk and CS activity that were associated with important aspects of lick kinematics. P cell suppression was associated with advancement of the tongue during both protrusion and retraction phases; SS increases were like associated with slowing the tongue during protrusion, likely aiding in endpoint accuracy. This appears to mesh well with past work in primate eye and mouse limb.

Recommendations for improvement

1) A fuller characterization of lick kinematics (lick angle, speed, etc.) and performance (tube miss rate) for different lick types (types 1 - 5 in Fig 1a) would be helpful

2) With respect to lick types 2 and 4, how do the authors separate licks precisely aimed to the edge of the tube versus a missed lick (e.g., if lick types 1 and 5 missed the spout)? 

3) An image or video of 'task irrelevant' licks, along with each lick type, would be helpful.

4) How were protraction and retraction defined? By speed minima, maximum lateral displacement, or some other metric? Perhaps this information is buried in the methods but I could not find it - this is an important issue to clarify key aspects of licking and I recommend clarification of this issue reside in the main text. 

5) What sensory modalities can marmosets use to find the food tube? Can the animals see the food tube, or purely by touch/miss information? Were the eyes tracked during food acquisition as well as during the saccade task? 

6) Are the authors missing a panel in Fig. 2 or is there an extra figure legend? There are only 4 figure panels (A -D), but 5 figure legends (A - E). 

7) Given that the tube miss rate was not described, I would guess that on some occasions, animals would miss the tube in a task-relevant lick, perhaps during CS suppression conditions described in Fig 3. Did the CS responses differ depending on hits vs. misses of the tube around the time of touch? If so, did the CS responses differ by the magnitude of the miss or by where the tongue contacted the tube? Similarly, did the magnitude of CS response depend on lick type? 

While the control experiments get at this point, I am specifically curious about endpoint error instead of kinematics preceding the CS event. 

8) Was the distance of the tongue tip to the mouth measured linearly from the tongue tip at max displacement to the mouth, or the curved distance defining the length of the tongue? Was lick tortuosity affected during any of these CS suppression conditions in Fig 3? More details on the defined aspects of lick kinematics would be helpful. 

9) Recent studies in mice have implicated cortical (Bollu, Ito et al., 2021, Nature; Xu et al, 2022, Nature) and collicular (Rossi et al, 2016; Li and Sabatini, 2021; Ito, Gao et al., 2024, bioRxiv) circuits for tongue control, and these studies are not cited in the paper. These studies describe both premotor signals and signals for errors in the motor cortex (missing a water spout) as well as unexpected touch events on the tongue. 

While the authors included a brief paragraph in the Discussion regarding multi-regional interactions among the motor cortex, superior colliculus and cerebellum, a paragraph discussing 1) whether signals similar to those reported in the two papers described above are predicted to be observed in marmosets, and 2) if so, how these signals could be integrated with the cerebellar signals described in the current manuscript, would be helpful in integrating results discovered across species in the manuscript. 

Minor

Time lag between SS rates between ipsi vs. contra. Is this a feature also observed in SS rates during saccades in PCs?

Did CS activity across PCs (Fig 2c) differ depending on contra vs. ipsi licks? 

Figure image quality was not preserved upon insertion, making them blurry when zooming in. This is particularly noticeable in Fig 3 when looking at the spike rates and kinematic changes. 

Fig. 3 took me a while to understand. For the heatmaps in Figs 3a, d and g, it would be helpful to clearly define that the plots show the difference between suppressed and non-suppressed (e.g., Dist. to mouth (Δ supp./not supp.)) within the figure itself.

It would be helpful for all panels in Figs. 3b, e, and h, to have the color of the single 'n' in 'n - (n -1)' yellow as in the top left panel of each figure for the reader.

---

## [Decision Letter · Decision Letter 2]

21 Feb 2025

Dear Reza,

Thank you for your patience while we considered your revised manuscript "Control of tongue movements by the Purkinje cells of the cerebellum" for publication as a Research Article at PLOS Biology. This revised version of your manuscript has been evaluated by the PLOS Biology editors, the Academic Editor and the original reviewers.

Based on the reviews, we are likely to accept this manuscript for publication, provided you satisfactorily address the remaining points raised by the reviewers, including the discussion of the papers mentioned by the reviewers insofar you think they are appropriate to include. Please also make sure to address the following data and other policy-related requests:

* We would like to suggest a different title to improve its accessibility for our broad audience: 

Purkinje cells of the cerebellum control tongue movements

* Please add the links to the funding agencies in the Financial Disclosure statement in the manuscript details.

* DATA POLICY:

Regardless of the method selected, please ensure that you provide the individual numerical values that underlie the summary data displayed in the following figure panels as they are essential for readers to assess your analysis and to reproduce it: 1E, 2BC, 3CF, S2B, S3C and S7.

* CODE POLICY

We expect to receive your revised manuscript within two weeks. 

*Published Peer Review History*

*Press*

Sincerely,

Christian

Christian Schnell, PhD

Senior Editor

cschnell@plos.org

PLOS Biology

Reviewer remarks:

Reviewer #2: In this revision, my previous major concerns are all addressed with additional data and analysis. However, I found that some of these additions are not fully integrated. In particular, even though they have demonstrated that non-CS SS pauses are sufficient to produce hypermetria just as CS-driven SS pauses, the abstract (lines 35-36) still describes the phenomenon as if it is specific to climbing fiber inputs (and therefore CSs). A similar omission can be seen at the beginning of the Discussion (lines 398-400).

In addition, I previously provided some references about non-CS SS pauses (Shin and De Schutter, J Neurophys, 2006; Hong et al., eLife 2015; Brown et al., Neuron 2024, etc.), but was surprised to find that none of them is cited in this revision. I am not claiming that the authors are obliged to cite those works, but wouldn't it be nice to acknowledge prior work on SS pauses, rather than completely ignoring them altogether? Also, given their explanation referring to molecular layer interneurons, I think that Lee et al., "Circuit mechanisms underlying motor memory formation in the cerebellum", Neuron, 2015 is very relevant. I hope that the authors take my suggestion seriously.

Reviewer #3 (Jesse Goldberg): This is a thorough revision that addressed our concerns, and I think that the response to reviewer 2 was substantial. The association of CS with errors is new and interesting in this revision. Overall this is a fantastic paper - a novel behavioral paradigm, precise kinematic tracking, and identification of important cerebellar signals for tongue control.

I have only one comment - likely easy to address - with citations in a revised discussion paragraph. I paste it below:

A cortico-cerebellar network for control of the tongue 

As the subject prepared to initiate a licking bout, the P-cells exhibited a ramping activity. Previous reports have noted that during this period there is ramping activity in the tongue regions of the motor cortex (54), as well as in the fastigial (54) and the dentate nuclei (55). Inhibiting the motor cortex in mice prevents both the onset and the termination of the licking bout (56), suggesting that both are active processes that are cortically mediated. Inhibiting the fastigial during the ramping period disrupts planning of the movement and removes the direction selectivity that the motor cortical cells exhibit (54), while inhibiting the dentate disrupts the ramping activity in the motor cortex (55). Similarly, exciting the P-cells in the vermis abolishes the ramping activity in motor cortex during the delay period of a decision-making task (57). This implies that as one prepares to initiate a movement, the rising activity in the motor cortex is controlled via a loop through the cerebellum. 

There is a mis-citation here. Ref 56 and Bollu et al, 2021 (not cited) both show that licks can be initiated during orofacial motor cortex photoinhibition in mice. This should be fixed.

---

## [Editor Report · Decision Letter 3]

10 Mar 2025

Dear Reza,

Thank you for the submission of your revised Research Article "Purkinje cells of the cerebellum control deceleration of tongue movements" for publication in PLOS Biology. On behalf of my colleagues and the Academic Editor, Peter Thier, I am pleased to say that we can in principle accept your manuscript for publication, provided you address any remaining formatting and reporting issues. These will be detailed in an email you should receive within 2-3 business days from our colleagues in the journal operations team; no action is required from you until then. Please note that we will not be able to formally accept your manuscript and schedule it for publication until you have completed any requested changes.

PRESS

We frequently collaborate with press offices. If your institution or institutions have a press office, please notify them about your upcoming paper at this point, to enable them to help maximize its impact. If the press office is planning to promote your findings, we would be grateful if they could coordinate with biologypress@plos.org. If you have previously opted in to the early version process, we ask that you notify us immediately of any press plans so that we may opt out on your behalf.

Sincerely, 

Christian

Christian Schnell, PhD

Senior Editor

PLOS Biology

cschnell@plos.org